# Neurons in primary auditory cortex represent sound source location in a cue-invariant manner

Katherine C. Wood [1,2], Stephen M. Town [1] & Jennifer K. Bizley [1]

Auditory cortex is required for sound localisation, but how neural firing in auditory cortex underlies our perception of sound sources in space remains unclear. Specifically, whether neurons in auditory cortex represent spatial cues or an integrated representation of auditory space across cues is not known. Here, we measured the spatial receptive fields of neurons in primary auditory cortex (A1) while ferrets performed a relative localisation task. Manipulating the availability of binaural and spectral localisation cues had little impact on ferrets' performance, or on neural spatial tuning. A subpopulation of neurons encoded spatial position consistently across localisation cue type. Furthermore, neural firing pattern decoders outperformed two-channel model decoders using population activity. Together, these observations suggest that A1 encodes the location of sound sources, as opposed to spatial cue values.

[1] Ear Institute, University College London, 332 Gray's Inn Road, London WC1X 8EE, UK. [2] Present address: Department of Otorhinolaryngology: HNS, Department of Neuroscience, University of Pennsylvania, Philadelphia 19104 PA, USA. Correspondence and requests for materials should be addressed to K. C.W. (email: woodkath@pennmedicine.upenn.edu) or to J.K.B. (email: j.bizley@ucl.ac.uk)

Our ability to localise sounds is important for both survival and communication. Auditory cortex (AC) is required for sound localisation in many mammals, including primates, cats and ferrets[1–5], and neurons in AC are sensitive to the spatial location of sounds[6–8]. However, perceptual thresholds for localisation are typically far narrower than spatial tuning of neurons[9–12], posing the question of how such cortical activity is related to spatial perception.

Studies of spatial representations in AC have often focused on the encoding of acoustic cues that support sound localisation. These include binaural cues such as inter-aural timing and level differences (ITDs and ILDs, respectively) that govern estimations of azimuthal sound location[13,14], and monaural spectral cues arising from pinna shape[15–18], which provide information about stimulus elevation and resolve front–back ambiguities. While spatial cues can provide redundant information, accurate sound localisation in multisource or reverberant environments frequently requires integration across multiple cue types[19,20]. Indeed, binaural and spectral cues are combined within the inferior colliculus[21,22], potentially enabling cortical representations that are better characterised by sound location than particular acoustic features[23,24]. Whilst most, but not all, neurons in AC represent sound source location relative to the head (i.e., an egocentric representation, indicative of a cue-based representation[8]), it is unclear whether spatial modulation in auditory cortical neurons reflects tuning to specific localisation cues, or a cue-invariant integrated representation of source location. This is because most studies have not considered the effects of systematically manipulating the available localisation cues on coding of spatial location.

Sound location plays a critical role in the analysis of auditory scenes and the formation of auditory objects[25–27]. Object-based representations are often elucidated by employing stimulus competition: for example, two sounds presented from different locations can be fused together[28] or repel one another[29], depending on whether other factors promote grouping or segregation. Consistent with an object-based representation in AC, the presence of a competing sound source can dramatically sharpen the spatial tuning of auditory cortical neurons[30,31], and cortical activity is consistent with a fused location when two sources are presented that elicit a the percept of a single intermediary sound source[32].

Although it is unclear what auditory cortical neurons are representing about sound location, several models exist of how neurons and neural populations represent sound location: distributed or pattern-recognition models (also referred to as labelled-line models, but referred to as distributed models hereafter)[33–37] posit that heterogeneous spatial tuning exists, with different cells narrowly tuned to specific sound locations (or their underlying acoustic cues) across the azimuthal plane. In contrast, the two-channel model[38,39] posits that tuning of cells is broad and conserved across a small number of subpopulations of cells, with space represented by the relative summed activity of two or more subpopulations (defined either by the hemisphere of the brain in which cells were recorded, or the hemifield of space to which cells are tuned). Evidence for and against each model exists: in agreement with two-channel models, spatial tuning curves of neurons in AC are generally broad, with peaks in contralateral space[8,38,40]. Similarly, spatial tuning of voxels from functional imaging data is consistent with a two-channel representation[23,39]. In contrast, distributed models are supported by recordings from neurons in gerbil AC, where cells represent ITDs across all sound locations rather than just one hemifield of space[35]. However, some experimental evidence cannot be explained by either model; for instance, neither model can account for deficits in contralateral sound localisation observed during unilateral

inactivation of AC[10,11,41]. To make progress, neural activity must be recorded with high spatial and temporal precision to avoid averaging signals over large populations of neurons (which may artificially favour the two-channel model).

The goal of our study is to test whether neurons in AC represent spatial cue values or sound source location, and to determine how population activity represents sound location. Since behavioural-state impacts spatial tuning[42], we record from animals engaged in discriminating the azimuthal location of sounds, while varying the availability of localisation cues. We use the resulting neural spatial receptive fields to assess whether primary auditory cortex (A1) encodes 'space' or auditory cue values, and whether a distributed or two-channel model provides the best description of the observed data. We find that a sub-population of neurons has stable spatial receptive field properties across spatial cue type, and these neurons encode the location of auditory stimuli across cues. Furthermore, the peaks of the spatial receptive fields are distributed across the contralateral hemifield, consistent with a distributed architecture but, importantly, only the contralateral hemifield is represented within each A1 consistent with inactivation studies showing contralateral localisation deficits[11,41,43]. Distributed decoders outperform two-channel decoders providing further evidence for the encoding of the location of auditory objects in A1 as opposed to representation of spatial cues.

## Results

**Relative localisation with complete and restricted cues.** To understand the representation of space in A1, we engaged ferrets in a two-interval forced-choice task, in which they reported whether a target sound was presented to the left or right of a preceding reference (Fig. 1). Reference stimuli were presented from −75° to +75° in azimuth in 30° steps at 0° elevation, with subsequent target sounds occurring 30° to the left or right of the reference location (at ±75°, targets always moved towards the midline). Both reference and target sounds were 150 ms in duration, separated by 20 ms of silence. Acoustic stimuli were either broadband noise (BBN, containing complete binaural and spectral cues), low-pass filtered noise (LPN: <1 kHz, designed to contain only ITD information), bandpass filtered noise (BPN: 1/6th octave centred at 15 kHz, containing ILD information and eliminating fine-structure ITDs and most spectral cues), or high-pass filtered noise (HPN: >3 kHz, containing ILDs and spectral cues and eliminating fine-structure ITDs).

Across locations, ferrets were able to perform the task with each of the acoustic stimuli (binomial test against 50%, $p <$ 0.001 for all ferrets and stimulus conditions, Supplementary Table 1, Fig. 1c). For each ferret, performance differences with restricted cues were assessed by measuring the effect of stimulus type on trial outcome (logistic regression with interactions, $p <$ 0.05, Supplementary Tables 2–5). All ferrets performed significantly worse with bandpass than broadband stimuli (Ferret F1302 across condition performance difference: −11.6%, F1310: −7.5%, F1313: −6%, Supplementary Fig. 1). Performance of ferrets with other cue-restricted stimuli varied; two ferrets performed significantly worse with low-pass stimuli where cues were limited to ITDs, although the magnitude of the difference was very small (F1302: −3.6%, F1310: −3.5%). One ferret performed worse with high-pass stimuli, where there were no fine structure ITDs (F1302: −6.8%). Given that animals successfully performed the task across conditions, we predicted that any change in spatial tuning in A1 for BBN versus cue-restricted stimuli would be modest, and most marked for the BPN stimuli, which consistently elicited worse performance across ferrets.

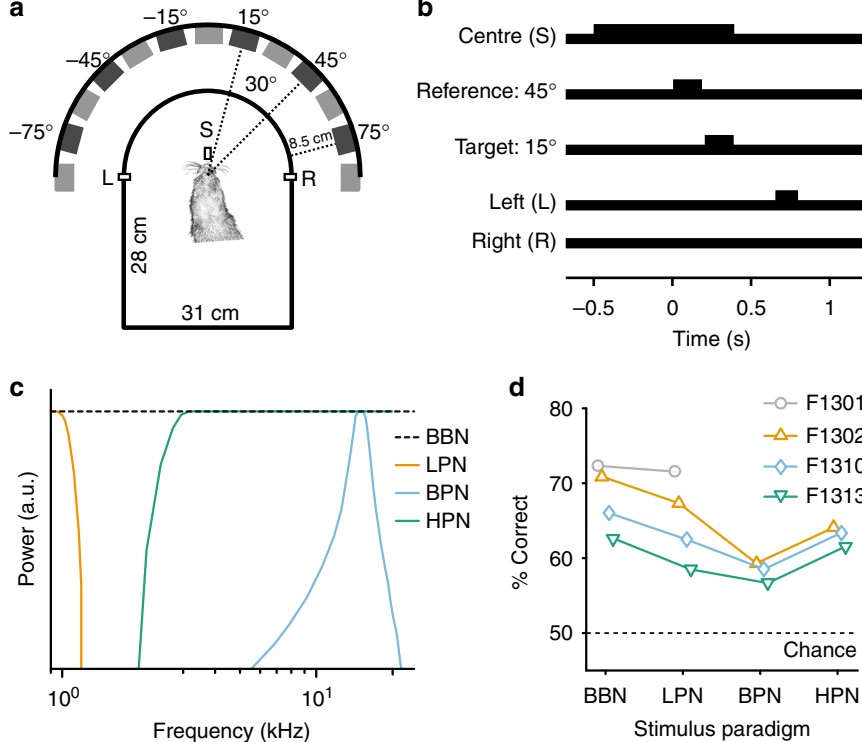

**Fig. 1** Relative localisation behaviour and performance. **a** Ferrets were trained to report the location of a target sound relative to a preceding reference, with target and reference sounds originating from two speakers separated by 30° (main speaker locations indicated). Ferrets reported the relative location of the target at a left (L) or right (R) response spout (positioned at ± 90°). **b** Ferrets were required to maintain contact with the central start spout (S) for a pre-stimulus hold time (500–1500 ms) before the reference sound (150 ms, shown here at 45°) was presented and remain there until at least halfway through the target sound (150 ms, here at 15°) before responding. **c** Schematic showing the spectrum of each of the stimulus paradigms. **d** Mean performance across all trials of ferrets in four different conditions, in which the stimuli were broadband noise (BBN), low-pass filtered noise (<1 kHz, LPN), bandpass filtered noise (1/6 octave about 15 kHz, BPN) and high-pass filtered noise (>3 kHz, HPN). All animals performed the task above chance in each condition (binomial test, $p < 0.001$)

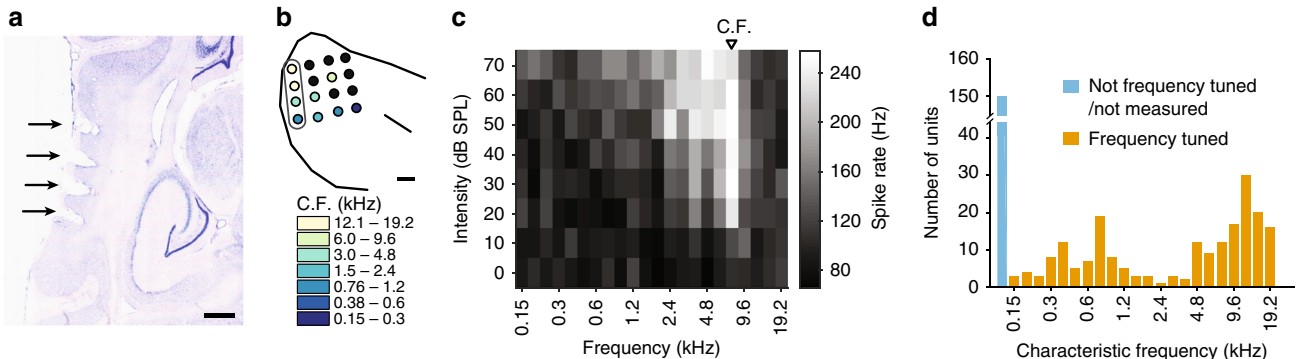

**Fig. 2** Experimental procedures. **a** Nissl-stained brain section of F1310 showing electrode tracks in A1 marked with electrolytic lesions (black arrows) of one column of the 4 × 4 electrode array. The locations of the electrode penetrations were clearly visible on the surface of the brain post-mortem, indicated in (**b**) with each circle representing the electrode sites. The circles are coloured according to the characteristic frequency (C.F., kHz) tuning of the units recorded from that electrode, assessed from frequency response areas (**c**) measured in response to pure tones. Black-filled circles indicates that no C.F. could be estimated for that electrode. The grey outline indicates the electrode tracks shown in (**a**). **c** Example, frequency response area showing a unit with C.F. of 7.6 kHz. **d** Distribution of characteristic frequencies for all recorded units. Scale bar in (**a**) and (**b**) indicates 900 μm

To assess changes in spatial tuning properties of neurons with differing spatial cue availability, we recorded the activity of 398 sound-responsive units from the left and right A1 while ferrets performed the task (total across all stimuli, ferrets, electrodes and recording depths, recording location confirmed with frequency tuning and post-mortem histology, Fig. 2). Spatial tuning was calculated during task performance by considering the neural response to the reference sounds only (Fig. 3).

**Spatial tuning in the auditory cortex of behaving ferrets**. To establish a benchmark of spatial tuning properties, we first characterised the responses to broadband stimuli containing a full complement of localisation cues. For each unit (e.g., Fig. 3a, b), we constructed a spatial receptive field using the firing rate across reference presentation (0–150 ms; Fig. 3c). We defined units as spatially tuned if firing rates were significantly modified by sound location (Table 1, Kruskal–Wallis $p < 0.05$). In total, 253 units

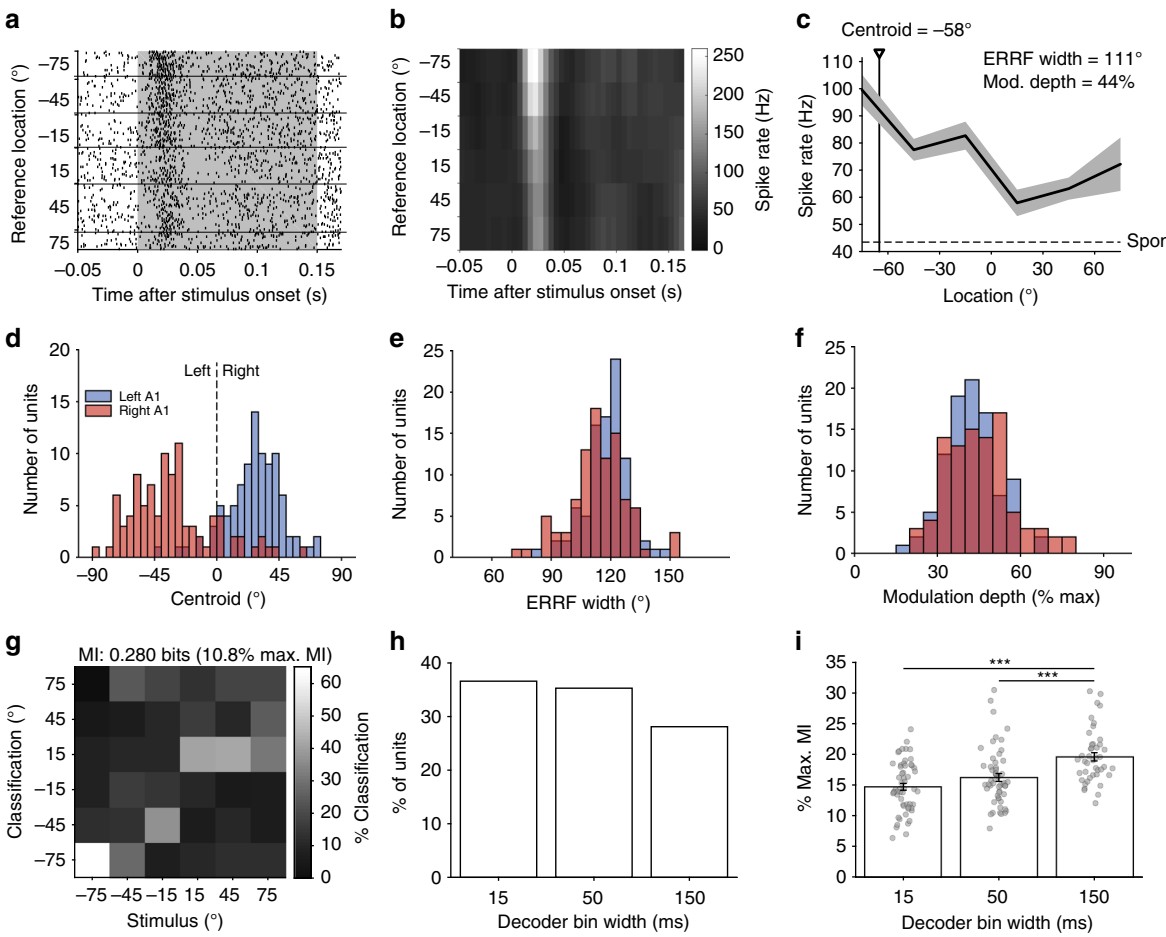

**Fig. 3** Spatial tuning properties in response to broadband noise. **a** Example, raster plot from one unit, with trials ordered by the location of the reference sound. The duration of stimulus presentation is indicated by the grey box. **b** Post-stimulus time histogram (spikes binned at 5 ms resolution). **c** Spatial receptive field (mean ± standard error of the mean (s.e.m.)) spike rate calculated over 150 ms at each location. The unit shows clear contralateral tuning and a centroid of −58° (black triangle/line). The equivalent rectangular receptive field (ERRF) width and modulation depth are also indicated in the figure. The dotted line represents the spontaneous firing rate. **d** Distribution of centroids, ERRF widths (**e**) and modulation depths (**f**) for all spatially modulated units (N = 190) split by left (blue, N = 95) and right (red, N = 95) auditory corttices (AC). **g** Stimulus location decoded with a Euclidean distance decoder on the spike response binned with 15 -ms resolution for the unit in **a**–**c**. The mutual information (MI) and percentage of maximum MI are indicated. **h** Distribution of best decoder bin widths for significantly informative units (N = 153). **i** Mean percentage of maximum MI ± s.e.m. of significant units from (**h**). Raw data are shown in grey circles. (N = 56, 54, 43). Statistics: one-way ANOVA with Tukey–Kramer post-hoc pairwise comparisons. ***p < 0.001

**Table 1 Spatial tuning properties of spatially modulated units**

| Stimulus | Total responsive units | Spatially modulated units (% of total) | Centroid (mean ± s.d.) | ERRF width (mean ± s.d.) | Modulation depth (mean ± s.d.) | Spatially informative units (% of total) |
|---|---|---|---|---|---|---|
| BBN | 253 | 190 (75%) | −30 ± 26° | 116 ± 13° | 44 ± 11% | 153 (60%) |
| LPN | 244 | 157 (64%) | −34 ± 22° | 119 ± 13° | 41 ± 13% | 112 (46%) |
| BPN | 97 | 63 (65%) | −28 ± 20° | 114 ± 10° | 44 ± 11% | 52 (54%) |
| HPN | 114 | 73 (64%) | −28 ± 29° | 120 ± 10° | 39 ± 9% | 47 (41%) |
| CSS | 128 | 98 (77%) | −31 ± 25° | 115 ± 12° | 46 ± 11% | 81 (63%) |

were responsive to BBN, of which 190 units (75%) were spatially tuned (Table 1).

For each spatially tuned unit, we described the preferred azimuthal direction by computing the centroid[31]. The majority of units (168/190, 88%) had contralateral centroids (mean ± standard deviation (s.d.) left hemisphere = 27.1 ± 20.2°, right hemisphere = −32.2 ± 30.1°, Fig. 3d). We also measured modulation depth and equivalent rectangular receptive field (ERRF) to determine tuning depth and width[42]. Tuning was generally broad (Fig. 3e, mean ± s.d. ERRF width, left hemisphere = 118.3 ± 10.7°,

right hemisphere = 113.9 ± 14.6°), with diverse modulation depths (Fig. 3f, mean ± s.d. modulation depth, left hemisphere = 42.5 ± 9.4%, right hemisphere = 43.8 ± 10.8%). There were no significant differences in the distributions of centroid values or modulation depths between the left and right hemispheres (unpaired T test, p > 0.05). However, there was a very small difference in the ERRF widths (the right hemisphere was narrower on average by 4.4°, unpaired T test, p = 0.019).

Spike timing conveys additional information about sound location beyond that offered by spike rates[44–46]. To test if our

units conveyed information about sound location in the temporal pattern of spikes, we decoded spatial location using a pattern classifier based on Euclidean distances between firing patterns in response to stimuli at each location, binned at 15, 50 or 150 -ms time resolutions (with 150 ms representing the firing rate across the whole stimulus). Classifier performance was summarised as the mutual information (MI) between actual and classified locations (Fig. 3g), and spatially informative units were identified as those with performance significantly greater than chance (permutation test, performance was deemed significant if the observed MI was more than the mean + two s.ds. of the MI calculated with shuffled sound locations, 250 iterations, effectively $p < 0.046$). We first used this measure to ask what proportion of units conveyed significant spatial information, and found that 60% (153/253) of units were spatially informative in at least one temporal resolution. When considering the best decoding window for each unit, all three decoding windows resulted in a similar proportion of spatially informative responses (logistic regression of bin width vs. constant model, $\chi^2 = 2.88$, $p = 0.0896$, d.f. = 1, Fig. 3h).

Whilst a similar proportion of units were spatially informative across all bin widths, the decoding approach allows us to quantify how much information is contained within individual responses. We therefore considered decoding performance, expressed as the percentage of the maximum available MI for perfect classification performance (where the maximum was defined as $\log_2(\#$ locations), see the Methods) and observed that performance was highest with the 150 -ms bin width (i.e., a spike rate code, Fig. 3i). Statistical comparison confirmed a main effect of bin width on decoder performance (one-way ANOVA, $F(2, 150) = 15.44$, $p < 0.001$) with post-hoc contrasts confirming greater MI for decoding with 150 ms than 15 ms (Tukey–Kramer corrected, $p < 0.001$) and 50 ms ($p = 0.001$) bin widths (Supplementary Fig. 2).

It is noteworthy that decoding performance was best when using neural activity across the whole stimulus window, suggesting sustained activity conveys information about stimulus location. We corroborated this finding by decoding spike rates calculated over increasing bin widths (50, 100 and 150 ms) and observing that significantly more units conveyed information at the longest bin widths (logistic regression, $\chi^2 = 72.1$, $p < 0.001$, d.f. = 1, Supplementary Fig. 3, 62% at 150 ms compared with 12% at 50 ms).

**A1 neurons represent space, rather than localisation cues**. We addressed the question of whether the auditory cortex represents space or localisation cue values by contrasting the spatial tuning observed in response to broadband and cue-restricted sounds. If an A1 neuron represents auditory space, then its tuning to sound location should be constant across different cues.

Individual units showed similar responses to cue-restricted and BBN stimuli (Supplementary Fig. 4A–C) and across cue-restricted stimuli (Fig. 4a–c). Centroids of spatially modulated units obtained in all cue-restricted conditions were neither different than those measured with BBN (KS test, $p > 0.05$, Supplementary Fig. 4D–F) nor between cue-restricted conditions (KS test, $p > 0.05$, Fig. 4d–f) across all units. When comparing only units spatially modulated in both conditions, there was a small change in the centroids between BBN and high-pass (mean ± s.d., 8.8 ± 2.7°, paired $T$ test, $p = 0.002$, Supplementary Fig. 4f), but no significant differences in the distributions of centroids for all cue-restricted condition pairs (paired $T$ test, $p > 0.0167$, Bonferroni-corrected alpha ($p = 0.0167$), Fig. 4d–f) or for BBN and low-pass or bandpass stimuli (Supplementary Fig. 4D, E). Thus, the direction of tuning was conserved across cue-restricted conditions.

In order to assess differences on an individual cell basis, non-parametric resampling of the data was performed to generate estimated confidence intervals on the centroid, ERRF width and modulation depth measurements (see the Methods section). This revealed that very few cells that were spatially modulated in pairs of cue-restricted conditions showed significant changes in centroid across stimuli (open circles, Fig. 4d–f, LPN-BPN: 3/22, LPN-HPN: 1/27, BPN-HPN: 2/23, Supplementary Fig. 5).

For ERRF width, we found significant differences when comparing units that were spatially modulated in either condition for broadband and high-pass ($T$ test, $p < 0.001$, Bonferroni-corrected alpha = 0.0167, Fig. 4g–i; Supplementary Fig. 4g–i) and for low-pass and bandpass ($p = 0.010$), but not for any other comparisons. However, when comparing units that were spatially modulated in both conditions, there was only a significant difference between broadband and high-pass stimuli (4.5 ± 1.2°, HPN > BBN, paired $T$ test, $p = 0.001$, black circles in Fig. 4g–i and Supplementary Fig. 4g–i).

Finally, for units spatially modulated in either condition, there were significant differences in the modulation depth between broadband and low-pass conditions ($T$ test, Bonferroni-corrected $p = 0.0125$), broadband and high-pass ($p < 0.001$), and low-pass and bandpass ($p = 0.003$, Fig. 4j–l; Supplementary Fig. 4j–l). When comparing units that were spatially modulated in pairs of conditions, there were again no significant differences in between any cue-restricted conditions (paired $T$ test, $p > 0.0167$, Fig. 4j–l). However, there were significant differences between broadband and low-pass stimuli ($-4.1 \pm 1.6\%$, $p = 0.012$), and broadband and high-pass stimuli ($-4.7 \pm 1.1\%$, $p < 0.001$, Supplementary Fig. 4j–l).

On a single-cell level, again very few units showed significant changes in ERRF width or modulation depth (Fig. 4g–l; Supplementary Figs 4, 5). Across the population of recorded neurons, the changes in tuning properties observed were of the same order of magnitude to those observed when we compared repeated recordings of the same stimuli for the same units (Supplementary Fig. 6).

Thus, the spatial tuning properties of individual units were maintained across stimuli that eliminated specific localisation cues. This suggests that A1 neurons can represent sound source location using spatial cues in a redundant fashion, consistent with a representation of space rather than individual (or specific) localisation cues.

**Representation of sound location across spatial cues**. We next asked whether reducing the available localisation cues impacted the ability to accurately decode sound source location from neural responses, and whether spatial information was conveyed over similar timescales.

Cue-type did not affect the optimal temporal resolution for decoding spatial information (Fig. 5a, logistic regression predicting unit classification (informative/uninformative) from bin width (15, 50 and 150 ms) and cue type: $\chi^2 = 11.5$, $p = 0.116$, d.f. = 4). Paired analyses for units tested in both BBN and cue-limited conditions revealed that the amount of spatial information (at the best bin width) in BBN and LPN conditions, and in BBN and BPN conditions was similar (Supplementary Fig. 7A, B, paired $T$-test, $p > 0.05$, $N = 62$ and 40, respectively). However, spatial information was significantly lower in HPN than BBN ($-5.2\%$, Supplementary Fig. 7C, paired $T$-test, $p < 0.001$, $N = 29$).

We also compared the best decoder performance of each unit (measured as the percentage of the maximum MI at the best bin width) across cue types (Fig. 5b). We found significant main effects of bin width (two-way ANOVA, $F(2, 352) = 21.83$, $p < 0.001$) and stimulus condition ($F(3, 352) = 5.16$, $p = 0.002$),

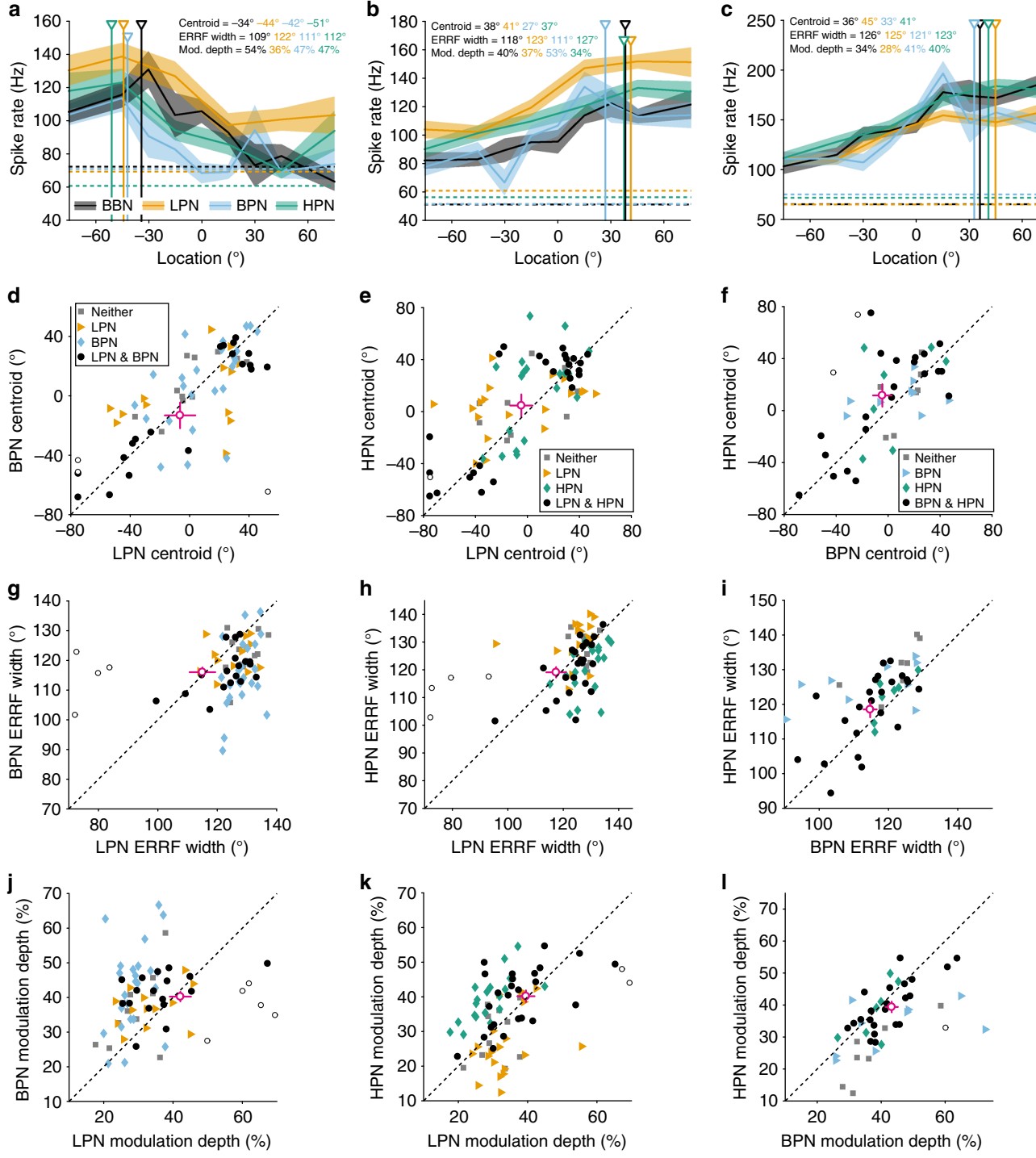

**Fig. 4** Comparison of spatial tuning properties between cue-restricted stimuli. **a–c** shows three example units recorded in all four cue conditions: broadband (BBN, grey), low-pass (LPN, orange), bandpass (BPN, blue) and high-pass (HPN, green). Centroid, ERRF width and modulation depth in each condition are indicated. The dotted lines represent the spontaneous firing rate. For panels **d–l**, units that were spatially tuned in both conditions (circles), either condition alone (diamonds or triangles) or tuned in neither (squares). Open circles indicate individual units that significantly changed between stimuli (non-parametric resampling, see Methods). Mean ± s.e.m. of the units spatially modulated in both conditions (i.e., of the black circles) is shown by cross-hairs (circle, magenta). **d–f** Centroids, ERRF width (**g–i**) and modulation depth (**j–l**) of units recorded in both LPN and BPN conditions (**d, g, j**; both spatially tuned N = 22, LPN tuned = 13, BPN tuned = 24, neither tuned = 11), LPN and HPN conditions (**e, h, k**; both spatially tuned N = 27, LPN tuned = 19, HPN tuned = 19, neither tuned = 10) and BPN and HPN conditions (**f, i, l**; both spatially tuned N = 23, BPN tuned = 9, HPN tuned = 7, neither tuned = 7)

but no interaction ($F(6, 352) = 0.13$, $p = 0.993$). Post-hoc tests (Tukey–Kramer, $p < 0.05$) were similar to the results with BBN stimuli (Fig. 3), with decoding being best when using a rate code (bin width of 150 ms). In addition, decoding of spatial location from responses to BBN was significantly

better than to BPN and HPN conditions (Tukey-Kramer, $p < 0.05$).

To elucidate whether units were representing the spatial location of sounds independently of their underlying spatial cues, we contrasted the number of units that were informative about

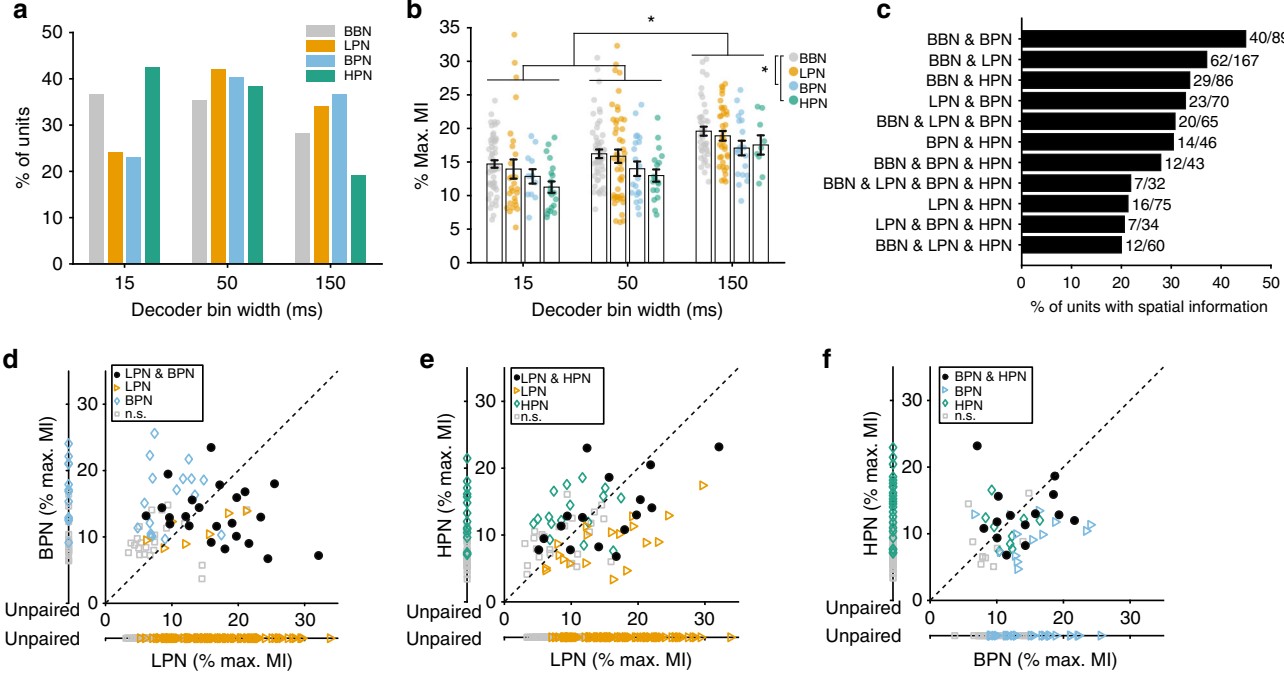

**Fig. 5** Comparison of spatial information in broadband and cue-restricted conditions. The amount of spatial information (MI) from stimuli with limited cues was compared with those where all cues were present (BBN) and between pairs of limited cue conditions. **a** Distribution of best decoder bin widths for all spatially informative units (BBN ($N = 153$), LPN (112), BPN (52), HPN (47)). **b** Mean percentage of maximum MI ± s.e.m. of units from (**a**). The raw data are shown by coloured circles. Statistics: two-way ANOVA with Tukey–Kramer post-hoc pairwise comparisons, asterisk (*) shows $p < 0.05$. **c** Proportion of units that showed information (in any bin width) in each combination of stimulus conditions. **d** Comparison of the mean percentage of maximum MI of units recorded in both low-pass (LPN) and bandpass (BPN. Paired: both significant $N = 23$, LPN sig. = 7, BPN sig. = 17, neither sig. = 23. Unpaired: LPN sig. = 82, not sig. = 92, BPN sig. = 12, not sig. = 15). **e** Low-pass and high-pass (HPN, paired: both significant $N = 16$, LPN sig. = 19, HPN sig. = 16, neither sig. = 24. Unpaired: LPN sig. = 77, not sig. = 92, HPN sig. = 15, not sig. = 24) and (**f**) BPN and HPN (paired: both significant $N = 14$, BPN sig. = 13, HPN sig. = 9, neither sig. = 10. Unpaired: BPN sig. = 25, not sig. = 26, HPN sig. = 24, not sig. = 44) conditions. Units that were only recorded in one condition are plotted along a separate axis. Units with significant MI in both conditions are shown as black circles, significant MI in only one of the conditions is indicated by triangles and diamonds. Grey squares show units not significantly informative in either condition

sound location across conditions in which distinct binaural cues were presented (i.e., LPN, containing ITDs, and either HPN or BPN, which did not contain fine-structure ITDs). We found that subpopulations of recorded cells were able to provide cue-independent spatial information: 33% (23/70) of units conveyed information about sound location across LPN and BPN and 21% (16/75) of units conveyed information across LPN and HPN (i.e., conditions with mutually exclusive cue types, Fig. 5c). For units that were informative in pairs of cue-limited conditions, there was also no significant difference in the amount of information (paired *T*-test, $p > 0.05$, Fig. 5d–f).

**Effects of a competing sound source.** If neurons encode auditory objects, rather than simply auditory space, then competition between objects might be expected to refine spatial tuning[25,29–31]. We hypothesised that a competing sound source would increase spatial sensitivity of neurons in A1 by causing competition between two objects. To test this, we repeated recordings in the presence of a competing sound source (CSS) consisting of a broadband noise presented at 0° azimuth and +90° elevation, whose level was randomly stepped within a range of ±1.5 dB SPL every 15 ms to aid segregation from the target stimuli.

Adding a competing source resulted in a mild impairment in relative localisation in one of three animals tested in this condition (F1310: mean change: −5.8%, Fig. 6a, GLM on response outcome with stimulus as predictor, $p = 0.011$, Supplementary Table 4). At the neural level, there was no effect of the competing source on the direction of spatial tuning in units

spatially modulated in both broadband and CSS conditions (Fig. 6b–e, paired T-test test, $p = 0.642$, $N = 59$) nor on the ERRF widths (Fig. 6f, paired *T*-test, $p = 0.301$). However, adding the competing source sharpened spatial tuning by increasing modulation depths ($p = 0.013$, Fig. 6g).

The competing source did not significantly impact decoding of sound location; neither the proportion of units with best decoding performance in each bin width (Fig. 6h, logistic regression: $\chi^2 = 4.98$, $p = 0.173$, d.f. = 2) nor the information content of units differed (Fig. 6i, j, two-way ANOVA, effect of CSS: $F(1, 233) = 0.29$, $p = 0.590$, Fig. 6i, no bin size-by-CSS interaction: $F(2, 233) = 0.63$, $p = 0.536$). Consistent with earlier results bin width also significantly affected decoding ($F(2, 233) = 15.35$, $p < 0.001$). Despite these similarities, population decoding was able to reach ceiling performance faster with BBN stimuli in the presence of a CSS than in its absence (see below).

**Decoding auditory space from population activity.** Spatially informative units conveyed on average between 15 and 19% of the maximum MI possible, implying that some form of population coding is necessary to reconstruct sound location perfectly. Reflecting the dominant theoretical descriptions of how neural circuits compute sound location, we applied three models of population decoding (Fig. 7a): (1) A distributed code[33,34] model (also referred to as a labelled-line code[35] or pattern code[36,37]) that decoded sound location from the activity pattern of neurons with heterogeneous spatial tuning, (2) a two-channel model that compared the summed activity of neurons in each hemisphere of

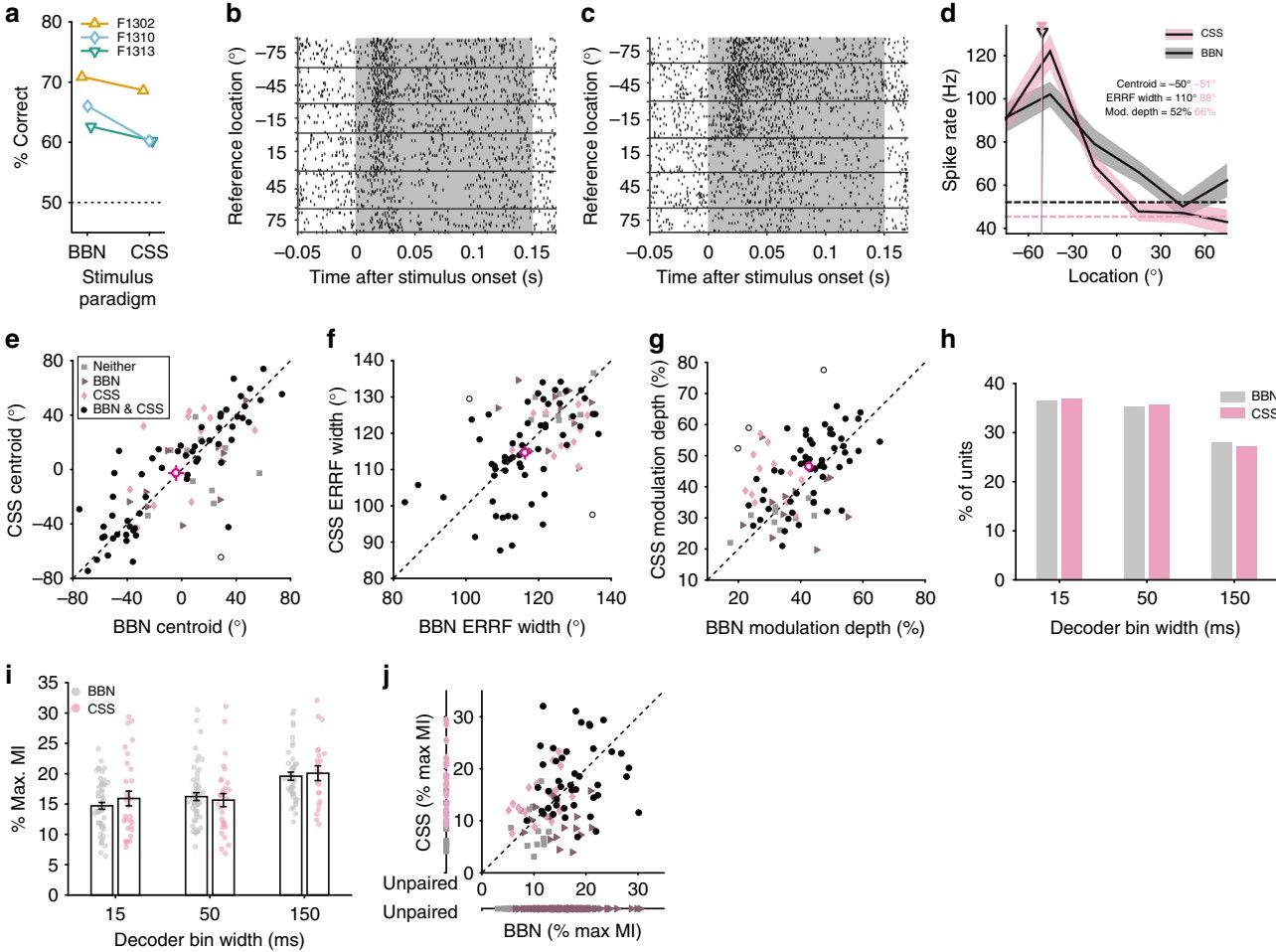

**Fig. 6** Spatial tuning properties in the presence of a competing sound source. **a** Performance in the relative localisation task in broadband conditions (BBN) and in the presence of a competing sound source (CSS) for each ferret. **b** Example, raster in response to BBN stimuli and (**c**) CSS stimuli. **d** Spatial receptive fields of the same unit with centroids indicated by triangles. Spontaneous firing rates are indicated by the dotted lines. **e** Centroids, (**f**) ERRF widths and (**g**) modulation depth of units recorded in both BBN and CSS conditions. Mean ± s.e.m. (standard error of the mean) of units significantly spatially modulated in both conditions (black circles, $N = 59$) is shown by magenta cross-hairs. Open circles show individual units that changed significantly between the BBN and CSS conditions. Units significantly spatially modulated in a single condition are shown by the triangles (BBN, $N = 11$) and diamonds (CSS, $N = 12$). Grey squares were units not significantly spatially modulated in either condition ($N = 10$). **h** Distribution of best bin widths for all significantly spatially informative units (BBN, $N = 153$; CSS, $N = 81$). **i** Mean maximum MI ± s.e.m. of units from (**g**). The raw data are shown in coloured circles. **j** Comparison of MI in units recorded in BBN and CSS conditions. Black circles indicate units with significant MI in both conditions ($N = 43$), units with significant MI in a single condition are shown by triangles (BBN, $N = 15$) and diamonds (CSS, $N = 18$). Grey squares did not have significant MI in either condition ($N = 16$). Units that were only recorded in one condition are plotted along a separate axis (labelled unpaired, BBN sig. $N = 95$, not sig. = 66. CSS sig. = 20, not sig. = 16)

the brain (hemispheric two-channel)[37,47,48], (3) a two-channel model that summed activity of two populations of neurons with centroids in left and right space, respectively (opponent two-channel)[38,39].

To assess whether recorded spatial receptive fields were more consistent with a distributed or two-channel code, we first compared predicted and observed distributions of correlation coefficients (R) calculated between tuning curves obtained for all pairs of units. The distributed model predicted a graded distribution of correlation coefficients, with many unit-pairs falling between −1 and 1 (Supplementary Fig. 8A)[33]. In contrast, the two-channel models both predicted distinct peaks in distributions at R = ±1, as tuning curves should be strongly correlated within (positive correlation) and between (negative correlation) channels (Supplementary Fig. 8B). We found that the distribution of correlation coefficients for units recorded in A1 most closely resembled the distributed model, with correlation

coefficients distributed broadly between +1 and −1 (Supplementary Fig. 8C–G). Thus, we predicted that the distributed decoder would outperform the two-channel decoders.

We next used maximum likelihood decoders (similar to ref. [35]) to estimate sound location on single trials using the joint distributions of spike rates (a) of individual units (distributed code) (b) across units from each hemisphere (hemispheric two-channel), or (c) across units tuned to each hemifield (opponent two channel). The distributed model decoder substantially out performed the hemispheric and opponent two-channel decoders in all stimulus conditions (Fig. 7b–f), with little difference in performance across stimulus conditions. The distributed decoder reached >85% correct with a minimum of 20 units (CSS condition) and maximum of 40 units (HPN), whereas neither of the two-channel models exceeded 45% correct. There was little difference between the performance of the opponent and hemispheric two-channel models, most likely because the

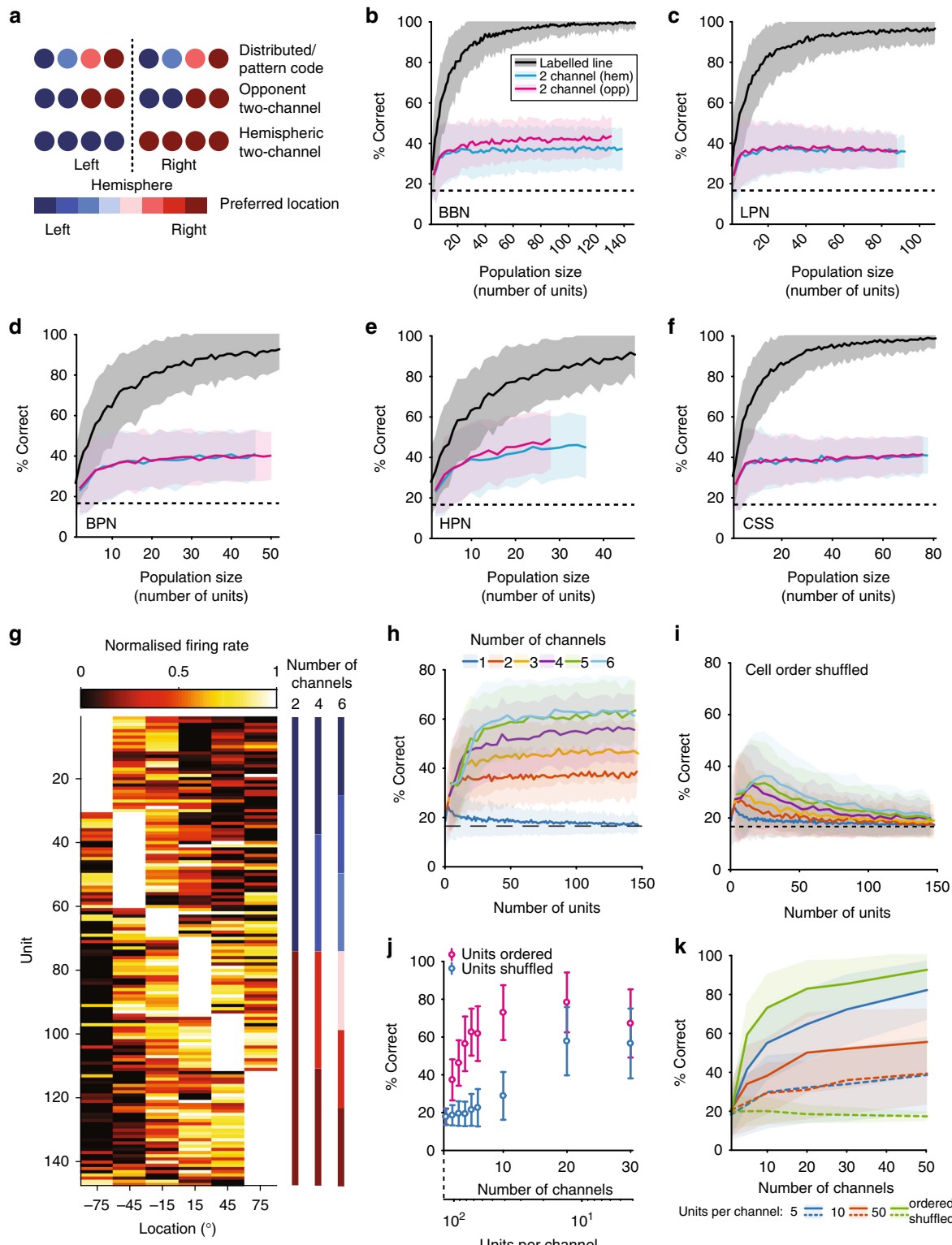

**Fig. 7** Distributed model population decoding outperforms two-channel models. **a** Schematic of three possible spatial coding mechanisms. **b–f** Performance of the two-channel (hemispheric–blue, opponent–magenta) and distributed (grey) decoders in each stimulus condition from populations of neurons that had significant MI about location. Performance of each decoder was calculated as percentage of correctly classified individual trials. The dotted line indicates chance (1/6). The black line and coloured bands indicate the mean ± s.d. of the performance for 250 different random subpopulations of each size drawn from the full sample of cells. **g** Spatial receptive fields for all units used in the population analysis for BBN, ordered by best position. **h** Decoder performance for models with increasing channel numbers with units grouped according to **g**. **i** Decoder performance when units are randomly ordered and assigned to channels. **j** Comparison of decoder performance and shuffled performance for a fixed population of 120 cells divided into increasing numbers of channels. **k** Effect of the number of units per channel using the data from modelled unit responses for ordered and shuffled units (see the Methods)

majority of neurons were tuned contralaterally (89% of units in response to BBN).

Previous work has suggested that spatial representations in the auditory cortex may comprise three channels[49,50], or many channels with 6° widths tiling space[51]. In order to investigate these ideas further, we modified our two-channel decoder to compare decoding performance of neural populations divided into $N$ channels according to spatial tuning observed in response to BBN stimuli. Cells were ordered by their preferred locations and divided into channels with equal numbers of units per channel (Fig. 7g). As the number of channels increased, decoding performance also increased (Fig. 7h). If the order of the units was randomly shuffled so that units within a channel were sampled without regard to their spatial tuning, performance was at chance, with the exception of populations containing very few units per channel (Fig. 7i). The fact that shuffled populations perform more poorly than the two-channel models suggests that the benefit of a system with many channels is not simply that it has more channels but that the labels of these channels are critical for spatial decoding.

When population size was held constant and the number of channels increased (such that there were decreasing numbers of units per channel, Fig. 7j, for a population of 120 units), shuffling the order in which units were grouped into channels prior to decoder training and testing always resulted in performance that was worse than ordered performance. However, as the number of channels increased and the number of units per channel decreased, the effect of shuffling unit order diminished. In the extreme case, where each channel had one unit, the shuffled and ordered distribution differed only in the relationship between channels and thus both shuffled and ordered populations provided distributed systems.

In order to further understand the relationship between the number of channels and the number of units per channel, we simulated responses of cells based on the spatial receptive fields of units that responded to BBN stimuli (units in Fig. 7g). This allowed us to investigate the effect of increasing the number of channels while keeping the number of units per channel constant (Fig. 7k). Increasing the number of channels rapidly improved decoding performance up to ~20 channels, after which performance saturated. For high channel counts (e.g., $N > 30$) and low numbers of units per channel ($n <= 5$), decoding performance of shuffled and ordered populations converged and performance levels were substantially lower. However, when the number of units per channel was high ($n = 50$), shuffling degraded performance from near ceiling to chance mirroring our observations in recorded neurons (Fig. 7j).

These results can be explained by a dependence of the distributed decoder on heterogeneity in the underlying tuning functions of individual cells and channels. When the number of units per channel was small and the number of channels was large, the diversity of spatial tuning curves present across units was preserved across spatial channels, and therefore the decoder remains successful in reconstructing sound location. This is true regardless of the order of units as, with only few units per channel, shuffling units and averaging within a channel still results in heterogeneity in tuning across channels. In contrast, when there are many units per channel, the order in which units are arranged before integration into channels is critical. When units are ordered by spatial tuning, adjacent units have similar spatial receptive fields and therefore the effect of integration is primarily to decrease noise, while channels remain strongly tuned to sound location, despite averaging. Thus, the heterogeneity of individual units is preserved at the channel level, while decoding performance improves due to reduction in noise. In contrast, when units are shuffled, adjacent units have differing spatial

receptive fields and so the effect of integration is to average out spatial tuning and make tuning very similar across channels.

Finally, if neurons represent spatial location, rather than merely spatial cue values, it should be possible to train a decoder with the responses to one class of stimuli and recover sound location when tested with the responses to a different class of stimuli. To test this, the normalised (z-scored) responses of neurons with significant spatial information in pairs of stimulus conditions were used to train and test the distributed model decoder across stimulus types. We trained the distributed decoder with responses to one condition (e.g., LPN) and tested the decoder using responses from the same neurons to another condition (e.g., HPN). To assess cue invariance, we chose the conditions that differed most clearly in the available binaural cues; the low-pass and high-pass stimuli, which contained only ITDs, or eliminated fine-structure ITDs to leave ILDs and spectral cues, respectively (Fig. 8). To quantify decoder performance, we considered both the % correct score and the unsigned error magnitude (mean observed RMS error, RMSE) and compared these values to that observed when the identity of the cells was shuffled prior to decoding to produce an estimate of chance performance (Fig. 8b). The likelihood of getting the observed difference in RMSE (or % correct) between the cross-cue decoder and the shuffled decoder was estimated by permuting the real and shuffled labels of the RMSE (or % correct) 1000 times. While decoding stimulus location with a decoder trained on the cross-cue condition resulted in worse performance than the within-cue condition (Fig. 8a), the error magnitudes were significantly smaller than chance (Fig. 8b) and the performance significantly greater than chance (% correct, Fig. 8c). The decrement in performance in the cross-cue condition was of the same order of magnitude (20–30%) as training and testing neural responses from the same units, to the same stimuli, but recorded on different days (Supplementary Fig. 9).

A comparison of low-pass (ITD) and bandpass stimuli (narrowband ILD) showed similar results (Supplementary Fig. 10), despite larger errors and lower performance for cross-cue testing, the results were still significantly better than chance. Interpreting the poorer generalisation from LPN to BPN stimuli is difficult: it might simply be that the decoding is done on a very small number of units and therefore relatively noisy (as can be seen in Supplementary Fig. 9). Alternatively, the narrowband nature of the bandpass condition renders it highly unnatural, and this manipulation had the largest effect on performance of the ferrets in the relative localisation task when compared to their performance with BBN (Fig. 1d). Therefore, it may be that the impaired behavioural performance is a consequence of the poor generalisation of cortical responses to this stimulus.

## Discussion

We sought to understand how the available localisation cues affect the encoding of azimuthal sound source location in the primary auditory cortex (A1). Reducing the available localisation cues (by eliminating ITD, ILD and/or spectral cues) had a weak effect on the animals' behaviour, and on the spatial tuning of neurons in A1. A subpopulation (20–30%) of units maintained their spatial tuning across conditions that contained contrasting binaural cues and were therefore capable of representing auditory space in a cue-invariant manner. Although tuning curves were predominantly contralateral, we found that units in each hemisphere had best azimuths that were distributed across the contralateral hemifield. Distributed and two-channel decoders, based on the spike rates of populations of neurons, both performed above chance in all stimulus conditions, but the distributed decoder outperformed the two-channel decoder. Together these

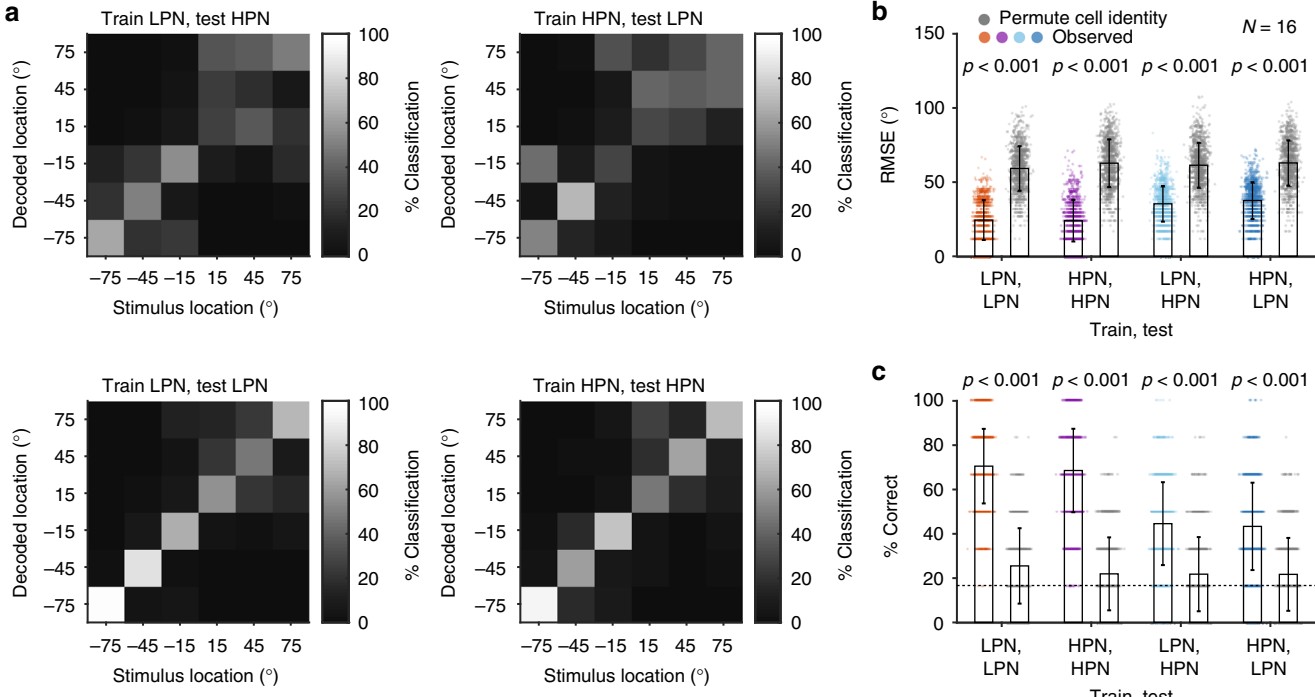

**Fig. 8** Performance of the distributed model decoder across different spatial cues. **a** Confusion matrices showing the % of decoded location classifications of each stimulus location when the distributed decoder was trained with responses to low-pass (LPN) stimuli and tested with responses to the high-pass (HPN) stimuli from the same units that had significant spatial information in at least one bin width (top left), decoder trained on HPN responses and tested on LPN responses (top right), trained and tested on LPN (lower left) and trained and tested on HPN (lower right). **b** Root-mean squared error (RMSE) of the decoder (coloured symbols) was lower than when the cell identities were shuffled (grey) in each condition tested. The bars show the mean ± s.d. across 1000 iterations for observed and cell-identity shuffled decoding. *P*-values indicate the probability that the difference between the observed and shuffled mean RMSE is greater than the difference obtained by chance (estimated by permuting the labels between real and shuffled decoder scores (1000 iterations) and calculating the difference in decoding, see Methods). **c** Mean ± s.d. % correct of the decoder (filled bars) compared with % correct when cell identity was shuffled (grey bars). *P*-values indicate the probability that the difference between the actual and shuffled mean % correct would be observed by chance. The grey dotted line indicates chance performance (1/6 chance of correct classification)

results suggest that auditory space is represented in an across-hemisphere distributed code, where spatial representation is divided across hemispheres, rather than each hemisphere containing a complete representation of space.

Ferrets were able to perform the relative localisation task using either ILD or ITD cues as evidenced by performance when cues were restricted to ITDs (by a low-pass noise stimulus), mainly ILDs (bandpass noise) or ILDs and spectral cues (high-pass noise). Many units only conveyed spatial information in the presence of either ITDs or ILDs. However, the directional preference of cells that responded in both conditions (as measured by centroids) was stable, and we identified a subset of roughly a quarter of units that were significantly spatially informative when provided with either ILDs or ITDs. Moreover, decoders trained on the neural responses to low-pass stimuli were able to recover location from the responses to high-pass stimuli and vice versa (Fig. 8). Together, these results suggest that a subpopulation of neurons in A1 provides a cue-invariant representation of sound location. The population of neurons that we were able to record across multiple stimuli was small (e.g., 16 units for low-pass and high-pass noise). While these small populations represented sound location with near-perfect accuracy when trained and tested with the same stimulus type, performance declined in the cross-cue condition. Since this decline was of a similar magnitude to decoders trained and tested with repeated independent recordings to the same stimuli, it seems likely that with a larger population of jointly sensitive neurons, performance could be maintained across multiple stimulus types. The way in which

our animals were trained may have minimised the possibility of observing cue-invariant neurons: in order to gain sufficient repetitions at each location, the stimulus type was fixed in a given testing session. If we were to interleave stimuli containing contrasting cues, we may have observed a greater number of neurons that responded in multiple conditions.

The evidence we observe for cue-invariant neurons parallels similar observations obtained using human neuroimaging, where regions of cue-independent and cue-specific voxels were observed[23]. These neuroimaging results suggest that a representation of azimuthal space exists within AC that is independent of its underlying acoustic cues. Our results provide the first cellular resolution evidence from the AC of behaving subjects in support of this hypothesis. It will now be important to ask how binaural cues are integrated within A1, and, given recent observations of world-centred spatial tuning in AC[8], in what reference frame cue-invariant neurons operate. Since many spatially tuned units were informative about sound location only when specific cues (ILDs or ITDs) were present, it is possible that the responses of cue-specific units are integrated within A1 to develop these cue-tolerant responses.

We observed that a distributed decoder outperformed a two-channel decoder, and that decoding performance scaled with the number of spatial channels. Taken together with the observations that (a) centroids tiled auditory space and (b) spatial tuning curves were less correlated across units than predicted by a two-channel model, our data support a distributed model of spatial encoding. However, rather than space being fully represented in

both hemispheres (as found in the gerbil[38]), each hemifield of space was represented by a distributed system within the contralateral A1. This is consistent with demonstrations of contralateral tuning bias in anaesthetised, passively listening or behaving cats[40,52], and ferrets[8,46], and awake monkeys[40]. Here, we build on these findings, by suggesting that each hemisphere contains multiple channels tuned to locations in contralateral space—rather than only two channels that represent left or right space as suggested previously[38]. This type of spatial encoding can explain why unilateral inactivation of A1 selectively impairs localisation in the contralateral hemifield[10,11,41]. Several questions arise and must be answered in the future: does the representation of auditory space become increasingly cue-invariant in non-primary AC, and what are the effects of unilateral hearing loss on cortical representations of sound location (as distinct from localisation cues)?

We tested a new model based on the two-channel decoder with $N$ channels containing similarly tuned neurons. This design allowed us to address the possibility that there may be more than two channels[49–51], and assess the impact of within-channel averaging on model performance. Thus, we made a version of the population decoder that compared channels of small populations of similarly tuned units that were summed together. We found that, as the number of channels increased, decoder performance increased, lending further support to a distributed encoding of auditory space in A1, where populations of similarly tuned units form spatial channels. Importantly, shuffling the spatial tuning of units decreased decoder performance in all cases; although above chance performance was observed where heterogeneity between the channels was maintained by having very small numbers of units per channel. This was consistent with the idea that averaging heterogeneous spatial receptive fields leads to loss of information[36,37,48]. Increasing the number of channels past ~20 did not substantially improve population decoding performance, suggesting that there may be an upper limit on spatial resolution for absolute localisation of auditory stimuli in the cortex, as has been suggested in humans[51].

The addition of a competing sound source sharpened spatial receptive fields by increasing modulation depth, consistent with previous findings in songbirds[30] and cats[53]. We did not observe any change in azimuthal spatial tuning with addition of the competing sound source. The competing source was located at 0° azimuth, and +90° elevation. It is therefore possible that shifts in tuning occurred that we could not observe by recording azimuthal spatial receptive fields at a single elevation. However, the observation that centroid positions were stable is consistent with findings in cats where a centrally located masker did not alter spatial location tuning[53]. Despite the sharpening of spatial tuning, there was no overall difference in how well spatial location could be decoded from spiking responses in the presence of a competing sound source. Nevertheless, population responses reached a ceiling with a smaller number of units than for other conditions, suggesting that the sharpening of spatial receptive fields enables more precise population decoding (Fig. 7). It may be that the observed changes in tuning with addition of the competing source reflect the neural correlates of selective attention to the behaviourally relevant target sound sources[54]. If true, then blocking attention-mediated changes in spatial tuning should impair relative sound localisation in the presence of a competing sound.

A distributed encoding of auditory space is consistent with the formation of auditory objects in A1, and would allow distinct subpopulations of neurons to represent separate sound sources[30,31]. This would require that separate spatial cues associated with the same object are integrated to generate a channel-based representation of sound location, as observed in human behaviour[51]. This process might begin in the midbrain, where

there is evidence of integration of spectral cues and binaural cues[21,22,55]. Determining whether AC forms representations of individual discriminable auditory objects (or sources) will involve testing under more naturalistic listening scenarios, where there is variation across multiple orthogonal properties[23,56,57], and the use of multiple sound sources in the context of a behavioural task.

## Methods

**Animals**. All animal procedures were approved by the local ethical review committee (University College London and Royal Veterinary College London Animal Welfare and Ethics Review Boards) and performed under license from the UK Home Office (Project License 70/7267) in accordance with the Animal (Scientific Procedures) Act 1986. Four adult, female, pigmented ferrets (*Mustela putorius furo*) were used in this study, 1–3 years old. The weight and water consumption of all animals were measured throughout the experiment. Regular otoscopic examinations were made to ensure the cleanliness and health of ferrets' ears. Animals were maintained in groups of two or more ferrets in enriched housing conditions.

**Behavioural task**. The ferrets were trained to perform a relative localisation task based on that performed by humans in ref. [58]. Ferrets reported the location of a target sound relative to a preceding reference sound presented from a location either 30° to the left or right of the reference (Fig. 1b). The behavioural task was positively conditioned using water as a reward. During testing days, ferrets did not receive any water in their home cage, although, when necessary, water obtained during testing was supplemented to a daily minimum with wet food.

On each trial, ferrets nose poked at a central port to initiate stimulus presentation after a variable delay (500–1500 ms). Trial availability was indicated by a flashing LED (3 Hz) mounted outside the chamber, behind the plastic mesh that enclosed the chamber, ~15 cm from the floor. Following sound presentation, subjects indicated their decision by responding at a left or right reward spout. Responses were correct if the animal responded left when the target was to the left of the reference, and right when the target was to the right of the reference. Ferrets always received a water reward for correct responses from the response spout and received an additional reward from the start spout on 5% of trials. Incorrect responses resulted in a 7 s time out before the next trial could be initiated, and were followed by correction trials (which were excluded from analysis) in which the trial was repeated. All training and testing was fully automated with all sensor input, stimulus presentation and solenoids coordinated via TDT System III hardware (RX8; Tucker-Davis Technologies, Alachua, FL) and custom-written software running in Open Project (TDT Software) and MATLAB (MathWorks Inc., Natick, USA). Training stimuli were broadband noise with the sound levels on each trial roved (55, 58, or 61 dB SPL). To maximise the number of trials per location for neural recordings, once trained and implanted, ferrets were tested at a single sound level, although each animal was tested with training stimuli at the beginning of each week. An animal was considered 'trained' when they reached criterion performance of 65% correct on the training stimuli (chance performance was 50%). Once trained, ferrets were chronically implanted with electrode arrays and subsequently tested with cue-restricted sounds and competing sound sources.

**Stimuli and speaker array**. All stimuli were noise bursts generated afresh on each trial in Matlab at a sampling frequency of 48 kHz. Sound stimuli were presented from thirteen loud speakers (Visaton SC 5.9) positioned in a semicircle of 24 -cm radius around one end of the testing chamber (Fig. 1a). Speakers were evenly positioned from −90° to +90° at 15° intervals, approximately at the height of the ferret's head at the central start spout. Speakers were calibrated to produce a flat response from 200 Hz to 25 kHz when measured in an anechoic environment using a microphone (Brüel and Kjær 4191 condenser microphone). The microphone signal was passed to a TDT System 3 RX8 signal processor via a Brüel and Kjær 3110–003 measuring amplifier. Golay codes were presented through the speakers, and the spectrum was analysed and an inverse filter was constructed to flatten the spectrum[59]. All sounds were low-pass filtered below 22 kHz (FIR filter <22 kHz, 70 dB attenuation at 22.2 kHz) and with the inverse filters applied. All the speakers were matched for level using a microphone positioned upright at the level of the ferret head in the centre of the semicircle; correcting attenuations were applied to the stimuli before presentation.

In testing, stimuli were two 150 -ms broadband noise bursts filtered according to the testing block being performed, including a 5 -ms cosine envelope at onset and offset, presented sequentially from two speakers separated by 30° with a 20 -ms silent interval. Locations tested were −75° to 75° at 30° intervals (−75, −45, −15, 15, 45 and 75°), although in some sessions (BBN and BPN), additional speaker locations at −30°, 0° or 30° were included. In a small number of early recordings (~3% of sessions from F1301 and F1302, for BBN: 19/544 and LPN: 11/339 recording sessions), speakers spanning −90° to 90° at 30° intervals were tested, in these recordings, stimuli were also 200 ms of duration. For these recordings, spatial tuning properties and MI decoding was performed on the first 150 ms after stimulus onset. In population decoding, only neural responses from sounds presented at −75° to 75° at 30° intervals were evaluated.

For cue-restricted testing, subjects were presented with either broadband noise at a single sound level (BBN), low-pass noise (LPN) where the stimuli were low-pass filtered (finite-duration impulse response (FIR) filter with cut-off 1 kHz with 70 dB attenuation at 1.2 kHz), bandpass noise (BPN, filter width one-sixth octave around 15 kHz; 47 dB attenuation at 10 kHz and 55 dB attenuation at 20 kHz), or high-pass noise (HPN filter cut-off 3 kHz; 70 dB attenuation at 2 kHz). All stimuli were presented at 61 dB SPL when measured at the position of the ferret's head at the centre spout. Low-pass stimuli were designed to present the ferret with ITD cues only; bandpass stimuli to provide ILD cues with very limited spectral cues, at this narrowband frequency ferrets rely on ILD cues to localise sounds[14] and the high-pass stimuli to exclude fine-structure ITD cues, but maintain ILDs and spectral cues.

BBN stimuli were also presented with a competing sound source (CSS), which consisted of continuous broadband noise whose amplitude was stepped within a 1.5 dB range (randomly drawn from a uniform distribution of values with mean 55 dB SPL) every 15 ms. The CSS was presented from a single speaker, located directly above the centre of the test arena. Reference and Target stimuli were presented at varying levels (55, 58, 61 dB SPL) in CSS recording blocks.

Ferrets were tested twice daily, Monday to Friday, with each stimulus condition being tested in two sessions on the same day over a 2-week period. At the beginning of each week, ferrets were run on the training stimuli and only proceeded to testing when they reached criterion (65%).

**Frequency tuning.** To determine the frequency tuning of units, ferrets were placed in an alternative testing arena with two speakers located on the left and right (24 cm from the ferret) at head height. Ferrets were provided with a constant stream of water from a central spout and passively listened to stimuli. Speakers were calibrated and matched for level (as described above). Sounds of varying frequency (150 Hz to 20 kHz at 1/3 octave intervals) at varying level (0 dB to 70 dB SPL) were diotically presented while recordings from the auditory cortex were made.

**Microdrive implantation.** The microdrives comprised 16 individually moveable, high impedance (~2 MΩ), tungsten electrodes in a WARP-16 device (Neuralynx Inc., Bozeman, MT) in a 4 × 4 array (~900 -μm spacing, adapted from[60]). Microdrives were implanted into left and right auditory cortex during an aseptic surgery. Anaesthesia was induced by a single dose of a mixture of medetomidine (Domitor; 0.022 mg/kg/h; Pfizer) and ketamine (Ketaset; 5 mg/kg/h; Fort Dodge Animal Health). The ferret was intubated, placed on a ventilator (683 small animal ventilator; Harvard Apparatus) and ventilated with oxygen and isoflurane (1–3.5%) to maintain anaesthesia throughout surgery. A local anaesthetic (Marcaine, 0.5%) was injected under the skin where incisions were to be made. An incision was made along the midline of the ferret's head, and the connective tissue cut to free the skin from the underlying muscle. For each hemisphere, the posterior 2/3 of the temporal muscle was removed, exposing the dorsal and lateral parts of the skull. Two anchor/ground screw holes were drilled into the posterior medial part of the skull and self-tapping bone screws inserted. A craniotomy was made over the auditory cortex. The microdrive was put in place over A1 using a micromanipulator so that the bottom of the array contacted the dura. The microdrive was then retracted, the craniotomy filled with Silastic and the microdrive replaced before the Silastic set. The ground wires of each implant were wound around each other and wound around the ground screws. A protective well with screw-cap was secured in place around the array with dental cement. A metal bar with two nuts was placed in the centre of the head to provide a head-fixing device for later electrode movement. Further local anaesthetia (Marcaine, 0.5%) was injected around the wound margin before the ferret was allowed to recover from the surgery. Post-operatively, ferrets were given pain relief (buprenorphine, 0.01–0.03 mg/kg) for 3 to 5 days post surgery and prophylactic antibiotics (Amoxycare LA, 15 mg/kg) and anti-inflammatories (Loxicam, 0.05 mg/kg) for 5 days post surgery.

After surgery, electrodes were moved into the brain and subsequently descended by 100–150 μm, whenever the ferret had completed a full cycle of behavioural testing. In this manner, over the course of 1–2 years, recordings were made from each cortical layer in each ferret. Testing was complete when the electrodes had moved to a depth below the estimated maximal depth of the ferret auditory cortex (2 mm). These data, combined with estimates of frequency tuning made at each site, enabled an estimate of the location of each electrode in the auditory cortex (Fig. 1).

**Neuronal recording.** Signals were recorded, amplified up to 20,000 times and digitised at 25 kHz (PZ5, TDT). The data acquisition was performed using TDT System 3 hardware (RZ2), together with desktop computers running OpenProject software (TDT) and custom scripts written in MATLAB. Headstage cables were secured to custom-made posts which screwed onto the protective caps that housed the implants, allowing the ferret free movement within the chamber.

**Spike sorting.** The raw broadband voltage trace was filtered using an elliptical filter with bandwidth 300–5000 Hz (MATLAB). The filtered trace was processed to remove noise correlated across channels[61]. Spikes were detected using an amplitude threshold set automatically and were clustered into 'units' using algorithms adopted from Wave_Clus[62]. Clusters were manually checked post-sorting and

assigned as multiunit or single unit. Units were defined as single if they contained fewer than 1% of all inter-spike intervals within 1 ms and they had a consistent spike wave-form shape. Recording sessions performed at the same depth, and within 3 days, were combined and spike-sorted as if they were a single recording session. The spike shapes, rasters and PSTHs of these sessions were then checked manually for how well the recordings combined and were rejected if there was any inconsistency (e.g., if the spike shapes were different, or if the PSTHs were different (T test, p < 0.05)). Where spike shape etc. differed between sessions, sessions were then spike sorted separately. Since multiple recordings were made at the same site in the brain, if the recordings could not be combined, then the 'best' recording session for the particular site was taken to ensure the same unit was not included in the analysis multiple times. The 'best' recording here refers either to the recording with the highest number of trials (for spatial tuning properties; centroid, ERRF width and modulation depth), or the recording with the best MI (for comparison of MI in individual units and for population decoding).

**Spatial tuning features.** Clusters identified in the spike sorting (see Supplemental Methods) were deemed sound responsive if the spike rate in the 50 ms post-stimulus onset was significantly different from the baseline firing rate in the 50 ms preceding stimulus onset (paired T test, p < 0.05).

Spatial receptive fields were defined by calculating the mean spike count over the 150 ms reference sound presentation, expressed relative to mean baseline activity measured in the 150 ms prior to stimulus onset. A unit was defined as spatially tuned, if its response was significantly modulated by location (Kruskal–Wallis test, p < 0.05).

The spatial tuning of each unit was given by its centroid and the location of the peak firing rate (best azimuth). The centroid was calculated in a similar way to Middlebrooks and Bremen[31]: the peak rate range was determined as one or more contiguous stimulus locations that elicited spike rates within 75% of the unit's maximum rate, plus one location on either side of that range. The locations within the peak range were treated as vectors weighted by their corresponding spike rates. A vector sum was performed, and the direction of the resultant vector was taken as the centroid.

The breadth of spatial tuning of a unit was represented by the width of its equivalent rectangular receptive field (ERRF)[42], which corresponds to the width of a rectangle with a total area that equals the area under the spatial receptive field, and with a height equal to the peak firing rate. Modulation depth was defined as ((max response–min response)/max response) × 100.

To estimate confidence intervals for the centroid, ERRF width and modulation depth of each unit, the spike rates from each location were non-parametrically resampled with replacement 500 times, taking the same number of trials for each location. The parameters were subsequently calculated from the resampled data resulting in a distribution of 500 observations for each parameter. The centroid, ERRF width and modulation depth of each unit were then taken as the mean of each distribution with the 2.5th and 97.5th percentiles giving the confidence intervals for each parameter.

Differences between the centroids in each stimulus condition were tested with a two-sample Kolmogorov–Smirnov test (p < 0.05), when considering the same units in each condition a paired T test was used (p = 0.0167, Bonferroni corrected). ERRF widths and modulation depths in each of the stimulus conditions were tested with paired (p = 0.0167, Bonferroni corrected) or unpaired T tests (p < 0.05), depending whether the comparison was across all units or between the same units, respectively.

**Spike pattern decoding of individual units.** Responses were binned with 15 ms, 50 ms or 150 ms resolution across the duration of the reference sound presentation. Spike patterns were decoded using a leave-one-out cross-validation procedure that compared the PSTH on a single test trial to template PSTHs calculated as the mean across trials for each location being tested, with the test trial excluded from template generation[63]. Test trials were classified by location according to the lowest Euclidean distance between test PSTH and template PSTHs. Mutual information (MI) was calculated between the actual and decoded sound location to quantify decoder performance. To test for significance, bootstrap simulations (250 repeats with resampling) on shuffled data were performed. The MI was deemed significant if it was more than two standard deviations above the mean of the shuffled distribution. Since extra locations were occasionally tested (BBN and BPN; see the section Stimuli and speaker array), the percentage of the maximum possible MI ($\log_2$ of the number of locations tested) was calculated, so that decoder performance could be compared across conditions with different speaker numbers.

**Population decoding.** Similar to Belliveau et al.[35], a Bayesian maximum-likelihood decoder was implemented to test different models of location coding. Three models were tested for decoding sound location: The distributed, the two-channel hemispheric and the two-channel opponent model. For the distributed model, the probability of a firing rate for each neuron in the population (N), given stimulus Y, was calculated as the product of the probabilities of the firing rate of each cell (i):

$$p(\text{firing rate}_N | Y) = \Pi_{(i=1)}^{N} p(\text{firing rate}_i | Y) \qquad (1)$$

For the two-channel hemispheric model, two populations of neurons were defined by the hemisphere of the brain they were recorded from. For the opponent-coding model, two populations of neurons were defined by the side of space their centroid occupied. The likelihood term was calculated using the mean firing rate across all neurons in the population at each location. The joint probability of a set of firing rates in each hemisphere was calculated as the product of the probability of the mean firing in each hemisphere:

$$p\left(\text{firing rate}_{\text{left}}, \text{ firing rate}_{\text{right}}|Y\right) = p(\text{firing rate}_{\text{left}}|Y) * p\left(\text{firing rate}_{\text{right}}|Y\right) \tag{2}$$

For the modified distributed model, the whole population of neurons ordered by best azimuth was divided into channels (1–30 channels) with equal numbers of units. The likelihood term was calculated using the mean firing rate of all neurons in each channel at each location. The joint probability of a set of firing rates in each channel ($i$) was calculated as the product of the mean firing in each channel (for N channels):

$$p(\text{firing rate}_N|Y) = \Pi_{(i=1)}^N \, p(\text{firing rate}_i|Y) \tag{3}$$

Since the number of presentations of a stimulus in a given recording session was not necessarily equal, the probability of the firing rate of each neuron at each location was calculated using the maximum number of trials that would result in equal presentation probability (randomly selected without replacement). Thus, for the purposes of the population decoder, each location was presented with equal probability. Thus $p(\text{stimulus} \mid \text{firing rate}_Y) \propto p(\text{firing rate}_Y \mid \text{stimulus})$ and therefore, the likeliest source location was defined as the max $p(\text{firing rate} \mid \text{stimulus})$.

The models were tested for populations increasing in size from 1 to the maximum number of units in each stimulus condition, with units randomly drawn from all the available units with replacement (bootstrap resampling). For each population size, a single trial from each cell was selected for decoding. This process was repeated 250 times for each population size. For each unit of the population, the mean and standard deviation of the spike counts from trials at each azimuth was calculated. Any unit recordings with fewer than seven trials at any location were excluded from the population testing. Units were selected from recordings where the locations tested spanned −75° to 75° in 30° steps, and where at least one of the bin widths had significant MI in the spatial location decoding. Where multiple test sessions were made at the same recording site, the best recording was defined as the unit with the highest significant MI value.

For testing the number of channels with fixed numbers of units (Fig. 7j), linear interpolation between the nearest actual values was performed so that the same, maximum number of units could be compared. For testing the effect of channel number with constant units per channel (Fig. 7k), we fitted Gaussian tuning curves to the population of units that responded to BBN and fulfilled the above criteria. For each population size (i.e., the number of channels and number of units per channel necessary), Gaussian tuning curves were selected from the population (with replacement) and 100 trials at each location were generated by drawing spike rates from Gaussian distributions with mean and variance of the fitted curves at each location. For testing the distributed decoder across stimuli and between different recording sessions of the same cells, responses were normalised by z-scoring. HPN was not assessed across sessions since the number of units was too low (N = 3). For cross-cue decoding, performance was compared to the performance of a cell-identity shuffled decoder. We quantified the effect of permuting cell identity on decoding performance by taking the mean performance observed from recorded data and subtracting the mean performance when cell identity was shuffled before decoding ($10^3$ iterations). To estimate whether this effect was significantly larger than would arise by chance, we computed a null distribution of effect size by remeasuring the difference in decoding performance when the labels for performance values (observed or permuted) were randomized (1000 randomizations). If cell identity was not important for decoding performance, we would expect the observed effect of permuting cell identity to be no larger than if the effect was computed after randomised resampling. In contrast, we observed a large effect of shuffling cell identity in the recorded (but not randomised) data. We estimated a p-value for our observed results as the proportion of effect sizes resulting from randomized data that were larger than the observed effect.

**Reporting summary**. Further information on research design is available in the Nature Research Reporting Summary linked to this article.

## Data availability
The data sets generated during and/or analysed in this study are available from the corresponding authors on reasonable request. The data presented in all figures are available from figshare with the identifier 10.6084/m9.figshare.c.4455089.

## Code availability
Custom-written computer code for behavioural and neural data collection and analysis is available from the authors on request.

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

## Acknowledgements

This work was supported by a Wellcome Trust and Royal Society Sir Henry Dale Fellowship (J.K.B. WT098418MA), a UCL Excellence studentship (K.C.W.) and the BBSRC (BB/H016813/1 to J.K.B.).

## Author contributions

K.C.W. and J.K.B. designed the experiments and wrote the paper; K.C.W. S.M.T. and J.K. B were involved in data collection; K.C.W. analysed the data.

## Additional information

**Competing interests:** The authors declare no competing interests.

