## [Transparent Peer Review File · Nature Communications]

Reviewers' comments:

Reviewer #1 (Remarks to the Author):

Primary auditory cortex represents the location of auditory objects in a cue-invariant manner. Significant findings.

This study challenges a long standing "two-channel" theory for how cortex contributes to the ability to spatially locate sounds which is fundamental to normal attention and survival in all mammals. The study and report provides convincing and ample evidence supporting an alternative multi-channel or "labeled line" model for how output from auditory cortex could be decoded to support sound localization behavior. There are several novel findings and new principles as detailed in the review. The figures and text and report in general are excellent and I have no critical comments. It is technically superb and will no doubt be high impact. There is only a very minor suggestion to add one sentence of clarification to the Results.

The study is technically robust. Firstly, they validate the sound design for varied sound location cues, they demonstrate that animals can accurately discriminate two sequential sound locations provided sounds contain a broad band noise (BBN, ≥ 5 octaves) that allows for all binaural and monaural cues. In contrast, when the noise bursts span only 1/6 octave about 15 kHz sound localization performance declines due to availability of ILD cues alone (Fig. 1). Secondly, they record neural activity with high spatial and temporal precision avoiding large scale averaging and it does so in animals as they are actively discriminating sound location. Finally, they simulate how the neural responses may be decoded in order for animals to make the sound location discriminations and identify novel principles for this decoding.

Several novel and significant neural coding properties and principles are revealed with this robust approach. First, they find neural responses that are spatially discrete within the cortex form a distributed code for sound location as opposed to an abrupt bimodal distribution as predicted by the "two-channel" theory. Secondly, they find cortical neural responses have broad overlapped response "tuning" to sound location within a given acoustic hemifield. Moreover, the neural spatial tuning resolution increases when there are simultaneous competing non-attended sounds. The latter suggests a circuit or network mechanism for attention mediated improvements in behavioral performance. Thirdly, they simulate how the actual spike output from auditory cortical neurons could account for the behavioral performance levels of the animals. Here they reveal multiple interesting novel principles. First, they demonstrate that there is information (aka MI) for accurate decoding of sound location over multiple time-scales up to 150 ms (Fig. 3). Moreover, including a bin of up to 150ms increases the performance of their simulated decoder suggesting there are responses following sound onset that contribute MI to decoding sound location differences between the first and second pulse of sound in their task. Fourth, they find simulations of a labelled line (distributed) decoder significantly out performs simulations of two different types of two channel decoders (e.g. Fig. 7). This is strong evidence that a labelled line model for cortical coding of sound location is much more consistent with the neural response data, behavioral data and more plausible than the previously proposed two channel models. Finally, they show performance increases with the number and heterogeneity of discrete neural responses (aka "channels") included suggesting that robust decoding of sound location emerges from information drawn from many neurons with discrete and complementary tuning properties. Again, this is strong support for a what they call the "labeled line" model where more labeled lines (aka recording sites or channels) yields improved "coverage", decoding and ultimately simulated spatial discrimination performance.

Minor Suggestion

1. Results, Line 191 onward, "confirming greater MI for decoding with 150 ms than 15 ms..." Here

it is important to add a phrase or sentence explaining the significance in the Results especially if it is not elaborated on in the Discussion. It is likely not an accident that most robust timescale for integrating and accurately decoding sound location was 150 ms time bin (e.g. Fig. 3I) happens to be the same time-frame for the sound burst duration (aka, 150ms) suggests that part of the useful temporal code could reside in the sustained or terminating response of the A1 neurons. This is interesting because typically A1 has marked onset and weak terminating responses and in general short duration responses and yet in this task and with these sounds there seems to be significant spiking going on late after sound onset that contributes information to decoding sound location.

Reviewer #2 (Remarks to the Author):

This is a highly significant study addressing the representation of sound-source locations in the auditory cortex. The demonstration that single neurons show similar spatial sensitivity to sounds differing in their content of intensive or temporal cues indicates a cortical representation of locations rather than of just cues. The results also compellingly favor a labelled-line representation in contrast to 2-channel models that have been favored in studies that evaluated only the integrated activity of large neural populations (e.g., EEG and MEG). I have only a few specific comments intended to clarify the presentation.

Line 11: I think that what is meant is that neurons encoded CONSTANT spatial position. Otherwise, one might think that a neuron was sensitive to multiple spatial cues, but the cues could correspond to differing locations.

Lines 29-30: To say that neurons are "tuned" to spatial location implies that a neuron responds selectively to sounds from a limited range of locations. That conflicts with typical observations that ERRFs can be around 120 degrees. Something like "are sensitive to ..." would be more appropriate.

Lines 49-52: This sentence is difficult to parse. The beginning makes sense, but the clause after the colon doesn't.

Lines 61-62: "perceived as perceptually fused" is redundant.

Lines 78-80: "both ... cannot" implies that the combination of the two models fails to account for the deficit. "Neither the 2-channel nor the labelled line can" would be clearer. The comma in line 78 doesn't belong.

Line 124: "where there were no ITD cues": There are scientists who have worked long and hard on ITDs in the envelopes of high-frequency sounds. This should say something like "no fine-structure ITD cues".

Lines 185-188: Please explain "percentage of maximum available MI" more completely. As I understand it, there were always 6 locations, so perfect identification of those locations would show a mutual information of $\log_2(6) = 2.59$ bits. I assume that, in each case, you computed MI in bits, and then expressed that as percent of 2.59.

Fig. 3: Please provide the ERRF width and modulation depth of the example unit in panel C to help the reader appreciate the distributions in E and F. Also, the use of a vertical axis in C that starts at 55 Hz is misleading. Many readers, including myself on the first reading, would glance at that plot and guess that the modulation depth was nearly 100%, which is clearly at odds with panel F. Please plot the data relative to a baseline of 0, or indicate the spontaneous firing rate.

Line 203-204: "Mean maximum MI" should read "Mean PERCENTAGE of maximum MI". Also, the "mean of ... units from G" is incorrect. As I understand it, G represents only one unit, whereas panel I shows the mean across the sample of all units.

Line 273: Again, this should be "Mean PERCENTAGE of maximum MI".

Line 298-299: I find it surprising that 70 units responded well enough to both <1-kHz lowpass and 15-kHz narrowband that one could evaluate spatial sensitivity. One expects neurons in area A1 to be fairly sharply tuned for frequency. I assume that these 70 units had CFs near 15 kHz and also responded to sounds in the low-frequency tails of their frequency response areas (FRAs). Please comment on the FRAs of these neurons. What was the level of the low-frequency sounds relative to the thresholds at those frequencies?

Line 316, "the ferrets being most impaired": Please clarify that this refers to the behavioral results.

Line 396: Please clarify that 85% refers to % correct, not % of maximum MI.

Line 472: Please give a sense of the magnitude of the "subpopulation". One might read this as a few outliers (i.e., maybe 10%), whereas I think you mean the majority. In line 473, I would say "the behavior of THE other units" to indicate that you mean the balance of all units rather than just some other subpopulation.

Lines 476-477, "were narrower than would be expected": You never tell us what tuning width you would expect for a two-channel model. You only tested a range of 150 degrees, and the distribution of ERRFs was around 120 degrees, which one might think is even broader than one might expect for a 2-channel model. That is, 150 degrees divided into 2 channels seems like 75 degrees. Incidentally, how do you account for the 2 units in Fig. 3E showing an ERRF greater than 150 degrees?

Line 481, "using both ILD and ITD cues": This sounds like the ferrets needed both cues, whereas I think the point is that they could do the task with either cue alone.

Line 745: I assume that "55, 58, or 61 dB" should read dB SPL. Please clarify. It would be helpful to indicate the typical thresholds of cortical units for, at least, BBN in this free-field setup.

Reviewer #3 (Remarks to the Author):

The manuscript entitled "Primary auditory cortex represents the location of auditory objects in a cue-invariant manner" reports on the nature of spatial coding in the primary auditory cortex (A1) of the awake, behaving ferret. By training animals to perform a spatial discrimination paradigm during extracellular recording, the authors show that on the population average, neuronal firing was impacted only insignificantly by altering the availability of individual spatial cues (ITD, ILD and spectral cues). They furthermore test different models of neuronal space representation and show that a multi-channel pattern recognition model was superior in reconstructing sound location compared to a two channel hemispheric average model.

The study tackles a timely topic of sensory representation, and is one of only a few to do so in the actively localizing animal, which is highly commendable. Consequently, the data is very interesting to the field. It is generally well presented, even though I was slightly confused by the

nomenclature and description of the models. However, inconsistencies exist with regard to interpretation and conclusions drawn from the findings, which I have summarized below.

Behaviour and Fig. 1:

How does behavioural data look when excluding performance at $\pm 75^\circ$? I am worried that these lateral positions made it too easy for the animals.

To what extent did the task really require "attention to the azimuthal location" (line 100)? Based on the paradigm description and personal experience with such stimuli, the extremely short gap (20 ms) between the reference and the target renders the task essentially a directional movement detection (did it move leftward or rightward).

Neuronal data:

In my opinion, the data presented in Fig. S3 and S5 are much more informative with regard to "cue invariance" than current Figs. 4&5, as it allows assessing to what extent individual units are tuned similarly to ITDs and ILDs etc. Fig. 4 is inconclusive, as whatever cue a unit might be sensitive to in the band-limited noise is also available in the BBN.

Related to above, I am a bit sceptical about the use of the term "cue invariance" given that most of the presented data consists of multi units, and that the "invariance" refers to the population average (the centroid). Individual units seem to alter their tuning quite a bit (and even lose or gain sensitivity), so any cue invariance is really only apparent on the population average level. This concept of a population average of spatial sensitivity is, however, later dismissed (see comments to Fig. 7)

Fig. 7: I got confused with the description of the models. The "labelled line model" is simply a pattern recognizer, not a topographic representation of cue sensitivity (what the term refers to in regard to the brain stem detector neurons).

What are the different two-channel models? In Fig. 7A, the figure label states "contralateral labelled line" and two channel, but in the text (lines 369ff) two different two channel models are mentioned?

It has been described before (and is somewhat trivial) that a model with more individual channels outperforms the two-channel model. That's also effectively what Figs. 7J,K show... But if the differences in tuning between units matters, why were centroids used in Figs. 4 to 6 to support the claim of cue invariance? Individual cells do shift, and that would affect the pattern classifier if I am not mistaken?

Along this line of thought, it would be most interesting to see how the performances would change if models are trained in one condition (e.g. BNN) and tested with responses to a different condition. This analysis would test the extent of invariance of the code for each model.

Out of pure curiosity, I wonder if the authors can make any statement how the decoding performances relate to the discrimination ability of spatial positions, given that the behavioural task required spatial discrimination (directionality, specifically).

Minor:

Line 284: Reference to panel E precedes C and D and then is repeated on line 291.

Line 369: Two models are mentioned, but three are introduced (and shown in Fig. 7).

Line 390: Potentially incorrect reference? 38 maybe?

Line 813: Surplus "."

Line 897: What were the criteria to assign clusters as multi or single units?

Line 1008: Reference to non-existent panel.

We thank the reviewers for their positive comments and constructive suggestions. We have addressed each of the comments individually below.

Reviewers' comments:

Reviewer #1 (Remarks to the Author):

Primary auditory cortex represents the location of auditory objects in a cue-invariant manner.

Significant findings.

This study challenges a long standing "two-channel" theory for how cortex contributes to the ability to spatially locate sounds which is fundamental to normal attention and survival in all mammals. The study and report provides convincing and ample evidence supporting an alternative multi-channel or "labeled line" model for how output from auditory cortex could be decoded to support sound localization behavior. There are several novel findings and new principles as detailed in the review. The figures and text and report in general are excellent and I have no critical comments. It is technically superb and will no doubt be high impact. There is only a very minor suggestion to add one sentence of clarification to the Results.

The study is technically robust. Firstly, they validate the sound design for varied sound location cues, they demonstrate that animals can accurately discriminate two sequential sound locations provided sounds contain a broad band noise (BBN, ≥ 5 octaves) that allows for all binaural and monaural cues. In contrast, when the noise bursts span only 1/6 octave about 15 kHz sound localization performance declines due to availability of ILD cues alone (Fig. 1).

Secondly, they record neural activity with high spatial and temporal precision avoiding large scale averaging and it does so in animals as they are actively discriminating sound location. Finally, they simulate how the neural responses may be decoded in order for animals to make the sound location discriminations and identify novel principles for this decoding.

Several novel and significant neural coding properties and principles are revealed with this robust approach. First, they find neural responses that are spatially discrete within the cortex form a distributed code for sound location as opposed to an abrupt bimodal distribution as predicted by the "two-channel" theory. Secondly, they find cortical neural responses have broad overlapped response "tuning" to sound location within a given acoustic hemifield. Moreover, the neural spatial tuning resolution increases when there are simultaneous competing non-attended sounds. The latter suggests a circuit or network mechanism for attention mediated improvements in behavioral performance. Thirdly, they simulate how the actual spike output from auditory cortical neurons could account for the behavioral performance levels of the animals. Here they reveal multiple interesting novel principles. First, they demonstrate that there is information (aka MI) for accurate decoding of sound location over multiple time-scales up to 150 ms (Fig. 3). Moreover, including a bin of up

to 150ms increases the performance of their simulated decoder suggesting there are responses following sound onset that contribute MI to decoding sound location differences between the first and second pulse of sound in their task. Fourth, they find simulations of a labelled line (distributed) decoder significantly out performs simulations of two different types of two channel decoders (e.g. Fig. 7). This is strong evidence that a labelled line model for cortical coding of sound location is much more consistent with the neural response data, behavioral data and more plausible than the previously proposed two channel models. Finally, they show performance increases with the number and heterogeneity of discrete neural responses (aka "channels") included suggesting that robust decoding of sound location emerges from information drawn from many neurons with discrete and complementary tuning properties. Again, this is strong support for a what they call the "labeled line" model where more labeled lines (aka recording sites or channels) yields improved "coverage", decoding and ultimately simulated spatial discrimination performance.

Minor Suggestion

1. Results, Line 191 onward, "confirming greater MI for decoding with 150 ms than 15 ms..." Here it is important to add a phrase or sentence explaining the significance in the Results especially if it is not elaborated on in the Discussion. It is likely not an accident that most robust timescale for integrating and accurately decoding sound location was 150 ms time bin (e.g. Fig. 3I) happens to be the same time-frame for the sound burst duration (aka, 150m) suggests that part of the useful temporal code could reside in the sustained or terminating response of the A1 neurons. This is interesting because typically A1 has marked onset and weak terminating responses and in general short duration responses and yet in this task and with these sounds there seems to be significant spiking going on late after sound onset that contributes information to decoding sound location.

Thank you for the suggestion; we have further explored the data in order to substantiate this claim and have incorporated this into the paper. The data are presented in a new supplementary figure (Figure S3, see below), which compared decoding performance for rate-only codes with increasing bin widths, and we include the following discussion.

Line 203-210: The finding that decoding performance was best when using neural activity across the whole stimulus window was noteworthy because it suggests that information about stimulus location was present in the sustained activity of units. To investigate this further, we measured the performance of the decoder with increasing bin widths (50, 100 and 150 ms) from stimulus onset. For units with significant MI, we found that decoding performance was highest in the longest bin width (Supplementary Figure 3), providing further evidence that information is integrated over time and conveyed in the sustained firing rate of units.

Supplementary Figure 3: Effect of increasing bin width on number of units with spatial

information. (A) Shows the percentage of units with the best decoding performance in 50, 100 or 150 ms bin widths. (B) Percentage of maximum MI of units from (A) in each decoder bin width. Statistics: one-way ANOVA Tukey-Kramer post-hoc pairwise comparisons. * p<0.05.

Reviewer #2 (Remarks to the Author):

This is a highly significant study addressing the representation of sound-source locations in the auditory cortex. The demonstration that single neurons show similar spatial sensitivity to sounds differing in their content of intensive or temporal cues indicates a cortical representation of locations rather than of just cues. The results also compellingly favor a labelled-line representation in contrast to 2-channel models that have been favored in studies that evaluated only the integrated activity of large neural populations (e.g., EEG and MEG). I have only a few specific comments intended to clarify the presentation.

Line 11: I think that what is meant is that neurons encoded CONSTANT spatial position. Otherwise, one might think that a neuron was sensitive to multiple spatial cues, but the cues could correspond to differing locations.

Thanks for this suggestion to improve clarity: We have edited accordingly:

Line 11: A subpopulation of neurons encodes constant spatial position across localisation cue type.

Lines 29-30: To say that neurons are “tuned” to spatial location implies that a neuron responds selectively to sounds from a limited range of locations. That conflicts with typical observations that ERRFs can be around 120 degrees. Something like “are sensitive to ...” would be more appropriate.

Edited:

Line 19-21: Auditory cortex is required for sound localisation in many mammals, including primates, cats and ferrets¹⁻⁵, and neurons in auditory cortex are sensitive to the spatial location of sounds⁶⁻⁸.

Lines 49-52: This sentence is difficult to parse. The beginning makes sense, but the clause after the colon doesn't.

We have edited this and hope that it now makes more sense.

Lines 40-43: Sound location plays a critical role in the analysis of auditory scenes and the formation of auditory objects: an auditory object is a stable perceptual representation of a sound source allowing the source to both be identified and localised in space^{25,26}.

Lines 61-62: “perceived as perceptually fused” is redundant.

Deleted ‘perceptually’.

Lines 49-54: Consistent with an object-based representation in auditory cortex, the presence of a competing sound source can dramatically sharpen the spatial tuning of

auditory cortical neurons 30,31. Moreover, when two distinct sound sources are perceived as fused, auditory cortical activity is consistent with the location of the fused percept, i.e. at a position between the two physical sound sources 32.

Lines 78-80: “both ... cannot” implies that the combination of the two models fails to account for the deficit. “Neither the 2-channel nor the labelled line can” would be clearer. The comma in line 78 doesn’t belong.

Changed:

Lines 70-72: However, some experimental evidence cannot be explained by either model: for instance, neither model can account for deficits in contralateral sound localization observed during unilateral inactivation of auditory cortex ^{10,11,39}.

Line 124: “where there were no ITD cues”: There are scientists who have worked long and hard on ITDs in the envelopes of high-frequency sounds. This should say something like “no fine-structure ITD cues”.

Thank you for pointing this out, we have edited the text as you suggest.

Lines 120-121: One ferret performed worse in the high-pass condition where there were no fine-structure ITD cues (F1302: -6.8 %).

Lines 185-188: Please explain “percentage of maximum available MI” more completely. As I understand it, there were always 6 locations, so perfect identification of those locations would show a mutual information of $\log_2(6) = 2.59$ bits. I assume that, in each case, you computed MI in bits, and then expressed that as percent of 2.59.

The reviewer here is correct that in the majority of cases we had 6 locations. However, in a small number of cases we had 7 locations: these are detailed in “Stimuli and speaker array” in the methods section, lines 843-849 and also lines 841-843 which describe that extra locations ($[-30^\circ, 0^\circ, 30^\circ]$ were tested in addition to -75° to $+75^\circ$ in 30° steps) were tested in BBN and BPN conditions. For consistency we therefore defined the performance of the decoder relative to its maximum, which as the reviewer correctly notes is simply a consequence of the number of stimulus conditions. We have clarified this within the manuscript.

Lines 194-198: While the best bin width of the decoder was equally distributed across units, the decoding performance (expressed as the percentage of the maximum available MI for perfect classification performance, where the maximum was defined as $\log_2(\# \text{ locations})$, see methods) was highest when decoding sound location from responses binned in a single 150 ms interval (i.e. a spike rate code, Figure 3 I).

Fig. 3: Please provide the ERRF width and modulation depth of the example unit in panel C to help the reader appreciate the distributions in E and F. Also, the use of a

vertical axis in C that starts at 55 Hz is misleading. Many readers, including myself on the first reading, would glance at that plot and guess that the modulation depth was nearly 100%, which is clearly at odds with panel F. Please plot the data relative to a baseline of 0, or indicate the spontaneous firing rate.

We have edited our figures throughout to indicate spontaneous rate on all spatial tuning functions.

Figure 3 has additionally been edited so that the ERRF width, modulation depth and spontaneous firing rate of the example unit are indicated. Also included is the % of max MI in panel G.

Line 203-204: “Mean maximum MI” should read “Mean PERCENTAGE of maximum MI”. Also, the “mean of ... units from G” is incorrect. As I understand it, G represents only one unit, whereas panel I shows the mean across the sample of all units.

The figure legend has been corrected, thank you

Lines 223: (I) Mean percentage of maximum MI of significant units from (H) (N = 54, 58, 46).

Line 273: Again, this should be “Mean PERCENTAGE of maximum MI”.

This has been changed

Line 302-304: (D) Comparison of the mean percentage maximum MI of units recorded in both low-pass (LPN) and band-pass (BPN), (E) low-pass and high-pass (HPN) and (F) band-pass and high-pass conditions.

Line 298-299: I find it surprising that 70 units responded well enough to both <1-kHz lowpass and 15-kHz narrowband that one could evaluate spatial sensitivity. One expects neurons in area A1 to be fairly sharply tuned for frequency. I assume that these 70 units had CFs near 15 kHz and also responded to sounds in the low-frequency tails of their frequency response areas (FRAs). Please comment on the FRAs of these neurons. What was the level of the low-frequency sounds relative to the thresholds at those frequencies?

The frequency tuning properties of units that responded to both the low pass and narrowband stimuli were mostly tuned either to high frequencies, as the reviewer suggested, or broadly (/un) tuned. We have included an example of each type of FRA below to illustrate the two types for the reviewer. There were also a few units tuned to middle frequencies where the FRA encompassed both the narrowband and low pass frequency ranges. Stimuli were presented in testing at 61 dB SPL. 60% of units (42 / 70) responded significantly (Paired T-test, stimulus evoked rate compared to spontaneous $p < 0.05$) to frequencies below 1 kHz, and at 15 kHz, when presented at, or below, 60 dB SPL. It is possible that the other 40% of units respond only when the animal is engaged in a task or simply responded poorly to pure tones.

Line 316, “the ferrets being most impaired”: Please clarify that this refers to the behavioral results.

Added in the word behaviour and a reference to the behaviour figure:

Lines 350-353: This is consistent with the ferrets’ behaviour being most impaired on BPN stimuli (Figure 1 C) and suggests that, for both ferret behaviour and auditory cortical responses, ITDs may provide more information about the location of a stimulus than ILDs alone.

Line 396: Please clarify that 85% refers to % correct, not % of maximum MI.

Added in ‘correct’:

Lines 442-445: The labelled-line decoder reached > 85% correct with a minimum of 20 units (CSS condition) and maximum of 40 units (HPN), whereas neither of the two-channel models exceeded 45% correct.

Line 472: Please give a sense of the magnitude of the “subpopulation”. One might read this as a few outliers (i.e., maybe 10%), whereas I think you mean the majority. In line 473, I would say “the behavior of THE other units” to indicate that you mean the balance of all units rather than just some other subpopulation.

Added in parentheses the size of the sub-population and the word ‘the’ before ‘other units’:

Lines 533-535: A subpopulation (20-30%) of units was capable of representing auditory space in a cue-invariant manner, while the responses of the other units was more consistent with the encoding of specific localisation cues.

Lines 476-477, “were narrower than would be expected”: You never tell us what tuning width you would expect for a two-channel model. You only tested a range of 150 degrees, and the distribution of ERRFs was around 120 degrees, which one might think

is even broader than one might expect for a 2-channel model. That is, 150 degrees divided into 2 channels seems like 75 degrees. Incidentally, how do you account for the 2 units in Fig. 3E showing an ERRF greater than 150 degrees?

Thank you for pointing out this inconsistency. As the reviewer notes our ERRF measurements are broad: this is unsurprising given our high spontaneous firing rates (the ERRF is defined as the width under the tuning curve of a rectangle with a height equal to the peak firing rate and doesn't take into account the minimum / spontaneous firing rate). Since our two-channel model lacks any spontaneous firing rate, a direct comparison is inappropriate and not terribly informative. We have therefore removed the claims about tuning width and instead focused on the distribution of peak firing rates and population decoding results. The two units with ERRFs of >150 degrees are two examples where we recorded over a wider range of stimulus locations as detailed in the methods:

Line 848-853: Locations tested were -75° to 75° at 30° intervals (-75, -45, -15, 15, 45 and 75°), although in some sessions (BBN and BPN), additional speaker locations at -30°, 0° and 30° were included. In a small number of early recordings (~3% of sessions from F1301 and F1302, for BBN: 19/544 and LPN: 11/339 recording sessions) speakers spanning -90° to 90° at 30° intervals were tested.

Line 481, “using both ILD and ITD cues”: This sounds like the ferrets needed both cues, whereas I think the point is that they could do the task with either cue alone.

Changed to “either/or”:

Lines 545-548: Ferrets were able to perform the relative localization task using either ILD or ITD cues as evidenced by performance when cues were restricted to ITDs (low-pass noise), mainly ILDs (band-pass noise) or ILDs and spectral cues (high-pass noise).

Line 745: I assume that “55, 58, or 61 dB” should read dB SPL. Please clarify. It would be helpful to indicate the typical thresholds of cortical units for, at least, BBN in this free-field setup.

Added in SPL. Unfortunately we don't have any rate threshold functions with BBN for these units / in this setup so we are unable to indicate typical thresholds. Thank you for pointing out the missing SPL. The lack of any behavioural difference in performance across level conditions suggests that these sound levels are suprathreshold.

Lines 814-815: Training stimuli were broadband noise with the sound levels on each trial roved (55, 58, or 61 dB SPL).

Reviewer #3 (Remarks to the Author):

The manuscript entitled “Primary auditory cortex represents the location of auditory objects in a cue-invariant manner” reports on the nature of spatial coding in the primary auditory cortex (A1) of the awake, behaving ferret. By training animals to perform a spatial discrimination paradigm during extracellular recording, the authors show that on the population average, neuronal firing was impacted only insignificantly by altering the availability of individual spatial cues (ITD, ILD and spectral cues). They furthermore test different models of neuronal space representation and show that a multi-channel pattern recognition model was superior in reconstructing sound location compared to a two channel hemispheric average model.

The study tackles a timely topic of sensory representation, and is one of only a few to do so in the actively localizing animal, which is highly commendable. Consequently, the data is very interesting to the field. It is generally well presented, even though I was slightly confused by the nomenclature and description of the models. However, inconsistencies exist with regard to interpretation and conclusions drawn from the findings, which I have summarized below.

Behaviour and Fig. 1:

How does behavioural data look when excluding performance at $\pm 75^\circ$? I am worried that these lateral positions made it too easy for the animals.

Performance of ferrets was actually lower in the most extreme positions than in the rest of space (i.e. discriminating direction of movement from $\pm 75^\circ$ to $\pm 45^\circ$). Performance increases if these discriminations are removed (see figure below where these locations have been removed). This is consistent with minimum audible angle changes in humans where the minimum detectable change in stimulus location increases with distance from the midline (Mills, 1958) and with decreased sensitivity at more lateral locations for relative localization judgments in human listeners performing this task (Wood and Bizley 2015, Bizley et al., 2015). Below is the % correct data for positions excluding ± 75 .

To what extent did the task really require "attention to the azimuthal location" (line 100)? Based on the paradigm description and personal experience with such stimuli, the extremely short gap (20 ms) between the reference and the target renders the task essentially a directional movement detection (did it move leftward or rightward).

Although not in the present paper, we ran the ferrets on a task with a longer gap between the stimuli of 100 ms. We have shown below that, although there was a slight drop in performance overall (Data shown below), ferrets were still well above chance in this task, providing support for localization of at least the first sound.

For one ferret, we also constructed stimuli that smoothly transitioned between reference and target locations: for this ferret, performance dropped substantially in this condition arguing – anecdotally at least - in favour of them performing a relative localization task rather than a movement discrimination task. Nonetheless, all of the analysis in this paper is based on the neural response to the first sound: whichever way the animal goes on to solve the task the animal must pay attention to azimuthal location as the sounds can originate from anywhere in frontal azimuth and therefore all reference locations occur at positions to which the animal is paying attention. This is in contrast to e.g. the Lee and Middlebrooks paper, where animals waited for a target sound that occurred at a different elevation: In that study, it would have been advantageous to direct spatial attention towards target elevation, and away from reference sound locations that were used to map the spatial receptive fields.

We have not included these points in the discussion at the present time, but if the reviewer feels it preferable we can include them.

Neuronal data:

In my opinion, the data presented in Fig. S3 and S5 are much more informative with regard to "cue invariance" than current Figs. 4&5, as it allows assessing to what extent

individual units are tuned similarly to ITDs and ILDs etc. Fig. 4 is inconclusive, as whatever cue a unit might be sensitive to in the band-limited noise is also available in the BBN.

Thank you for this suggestion; on reflection we agree that the evidence presented in figures S3 and S5 support the main finding more than those presented in figures 4 and 5. We have swapped figure 4 and S3 directly. In figure 5, panels C-E have been swapped with the panels in S5

Related to above, I am a bit sceptical about the use of the term “cue invariance” given that most of the presented data consists of multi units, and that the “invariance” refers to the population average (the centroid). Individual units seem to alter their tuning quite a bit (and even lose or gain sensitivity), so any cue invariance is really only apparent on the population average level. This concept of a population average of spatial sensitivity is, however, later dismissed (see comments to Fig. 7)

Our definition of ‘invariance’ was that the unit had to convey spatial information across both high and low frequency cues (e.g. Fig 5C, units that are jointly sensitive to e.g. LPN and HPN or LPN and BPN, which vary between 20% and 30% of the units tested on each combination). These are the units that are shown with black symbols in Figure 4 (i.e. are informative about both cue types for each of the comparisons). Looking at the centroid plots in Figure 4, it is true that centroid values move around quite a lot for some units. However, the centroids of units shown in black, which we describe as invariant, are almost exclusively along the line of equality, indicating that their tuning in each condition is consistent. The majority (70%) of units were not spatially sensitive in both conditions – indicating that, in all likelihood, they encode either high frequency level or low frequency timing cues rather than the spatial location that would elicit those cue values. When spatial tuning is compared across full-cue (i.e. BBN, Fig.S3) and cue-restricted conditions, most centroids again lie along the equality line, indicating that their spatial position has not changed.

We have edited the manuscript to hopefully make this point clearer:

Lines 236-245: We first determined the impact of eliminating spatial cues by comparing the centroids of spatially modulated units to either broadband or limited cue stimuli (Supplementary Figure 4), before comparing tuning properties across different cue-restricted stimuli (Figure 4). Centroids obtained for all cue-limited conditions were no different to those measured with BBN (KS test $p > 0.05$, Supplementary Figure 4 D-F) nor between limited cue conditions (KS test, $p > 0.05$, Figure 4 D-F). The distribution of centroid differences was centred at zero in both cases, and the centroids of individual units were conserved across broadband and cue-restricted conditions (Supplementary Figure 4 D-F).

With respect to spatial averaging I think the reviewer may have misunderstood us here: Our population data shows that averaging across units improves decoder performance – but only if there are sufficient channels and the units within each channel share spatial tuning. The ‘best’ population decoder is one with multiple channels, where each is comprised of the average response to a number of similarly tuned units. Spatial averaging is therefore advantageous for population decoders constructed with multiple spatial channels (i.e. a labeled line) where the responses of similarly tuned neurons are averaged within each channel.

Fig. 7: I got confused with the description of the models. The “labelled line model” is simply a pattern recognizer, not a topographic representation of cue sensitivity (what the term refers to in regard to the brain stem detector neurons).

It is true that the labelled-line decoder is a pattern recognizer but in the auditory spatial localization literature this has become the more standard way to refer to this type of decoder (e.g. see Belliveau et al. 2013). In a sense it is ‘labelled’ since a specific population of neurons represents a particular location and the activity of those neurons is necessary to determine that a sound source is present at that location. We find no evidence for a topographic representation of auditory space other than that each hemifield of physical auditory space is represented in the contralateral primary auditory cortex

We have clarified this point where we first introduce the models:

Line 421-423: “Note that the labelled-line model is essentially a pattern code and there is no requirement for a topographic representation of space.”

What are the different two-channel models? In Fig. 7A, the figure label states “contralateral labelled line” and two channel, but in the text (lines 369ff) two different two channel models are mentioned?

We apologize for this as we now realise that the description of the models was somewhat confusing. The two-channel models tested were the:

(A) Hemispheric model, which compared the summed activity of neurons in left and right hemispheres of the brain

(B) Opponent model, which compared the summed activity of populations of neurons tuned to sounds in left and right hemifields of space (where hemifield of tuning was determined by the unit’s centroid).

We have edited the text to make the description clearer and also modified Figure 7A to reflect the three main models tested. The three models are explained in the main body of the text:

Lines 405-412: To address this we used three models of population decoding to reconstruct sound location from neural activity (Figure 7 A): (1) A labelled-line model that decoded sound location from the activity pattern of neurons with heterogeneous spatial tuning, and (2) a two-channel model that compared the summed activity of

neurons in each hemisphere of the brain (hemispheric two-channel) or (3) a two-channel model that summed activity of two populations of neurons with centroids in left and right space respectively (opponent two-channel).

It has been described before (and is somewhat trivial) that a model with more individual channels outperforms the two-channel model. That's also effectively what Figs. 7J,K show... But if the differences in tuning between units matters, why were centroids used in Figs. 4 to 6 to support the claim of cue invariance? Individual cells do shift, and that would affect the pattern classifier if I am not mistaken?

We found that for population decoding, a labeled line model out performed two-channel models and that performance increased with the number of channels in the model, so that a system with up to 20 channels, that averaged the responses of a small number of similarly tuned neurons. So, at the population level the differences in tuning across the neural population are important.

A separate point that we make earlier in the paper is that the centroids of individual neurons are constant across changes in the availability of acoustic cues supporting the idea that these neurons represent a spatial location rather than one of the spatial cues that might correspond to that location. We have addressed this later point above, and go on to explore this with respect to population decoding as suggested by the reviewer in the next point.

We apologize if we are misunderstanding something, but we aren't quite sure how the issues the reviewer raises are linked together...

The reviewer is correct that, by itself, the issue of channel number is relatively trivial, and it's unsurprising that 20 channel systems outperform 2 channel systems (although we note that Lesica et al. (2010) found that the two-channel model and the labelled-line decoders performed equally with data recorded from the gerbil IC). However, an important feature of our analysis is that we compare shuffled and spatially ordered systems with matched numbers of channels. The key take-home message from this analysis is therefore that the benefit of a many-channeled system isn't just that it has more channels, but the labels of these channels are critical for spatial decoding. If this were not true, then shuffled labelled line systems with many channels would perform better than two-channel models, simply as a bi-product of having more channels. This is not the case, with the exception of very high channel counts in which case, the system degenerates into a pattern classifier as the reviewer previously notes.

We added additional text to clarify this point: line 469-472:

That shuffled populations perform more poorly than the two-channel models suggests that the benefit of a system with many channels is not simply that it has more channels but that the labels of these channels are critical for spatial decoding.

With regard to the centroids, our analysis in Fig 4 and 6 simply shows that these are consistent for many units in the population across different conditions. This analysis is essentially thus a within-unit analysis, in contrast to the point we draw in the population decoding, where we emphasize the variation in centroids between units.

Indeed, the training of population decoders in one condition and test in another, as the reviewer suggests below, allows us to study the effect variation in centroids across conditions on population decoding. As shown below, population decoders generalize well across cues, indicating that variation in centroids does not affect classification accuracy.

Along this line of thought, it would be most interesting to see how the performances would change if models are trained in one condition (e.g. BBN) and tested with responses to a different condition. This analysis would test the extent of invariance of the code for each model.

We initially did not include this comparison because the number of units recorded in two conditions that also had significant MI in both conditions (a prerequisite for inclusion in the population decoder) was low. However, by employing a strategy also used in Belliveau et al. 2013, whereby the number of units is effectively doubled by assuming that each unit has a corresponding counterpart in the other brain hemisphere, we were able to perform this analysis: that is we trained a decoder with one set of spatial cues and determined to what extent we could decode spatial location using the responses to an alternate set of cues (e.g. train on ITDs test on ILDs and vice versa). We found that while performance dropped for cross-cue decoding, decoding was nonetheless always better than chance.

We added the following to the results (Lines 507-521), and a new supplemental figure 8:

Finally, if neurons represent spatial location, rather than merely spatial cue values, it should be possible to train a decoder with the responses to one class of stimuli and recover sound location when tested with the responses to a different class of stimuli. To test this, the responses of neurons with significant spatial information (MI) in pairs of stimulus conditions were used to train and test the population decoder across stimulus types. To obtain a large enough number of units to perform this analysis, we assumed that units would exist in the opposite brain hemisphere and have identical but mirror symmetric spatial tuning. We therefore effectively doubled the population of recorded cells by flipping the spatial receptive fields about the midline as in 38 and simulating responses (as in Figure 7 K). We trained the decoder with responses to one condition (e.g. BBN) and tested the decoder using responses from the same neurons to another condition (e.g. HPN). Figure S8 shows performance of the N-channel decoder when trained and tested across conditions. While performance declined for across-stimulus decoding, performance exceeded chance in all cases. If neurons represent spatial location rather than merely spatial cue values, it should be possible to train a decoder with the responses to one class of stimuli and recover sound location when tested with the responses to a different class of stimuli. To test this, the responses of neurons with significant spatial information (MI) in pairs of stimulus conditions were used to train and test the population decoder across stimulus types. To obtain a large enough number of units to perform this analysis, we assumed that units would exist in the opposite brain hemisphere that had identical, but mirror symmetric spatial tuning. We therefore effectively doubled the population of recorded cells by flipping the spatial receptive

fields about the midline and simulating responses (as in 38). We trained the decoder with responses to one paradigm and tested the decoder using responses from the same neurons to another paradigm. Figure S8 shows performance of the N channel decoder when trained and tested across stimuli. While performance declined for across-stimulus decoding, performance exceeded chance in all cases.

A section was added to the methods:

Lines 1093-1098: For the population decoding across paradigms (Supplementary Figure 8), to increase the number of units available for testing, the spatial receptive fields of all neurons were flipped around the midline effectively doubling the number of units available (a strategy used in 38). Since overall firing rate was different in response to different stimulus conditions, the spatial receptive fields were normalised to between 0 and 1, maintaining the spatial tuning relationship between stimuli.

Supplementary Figure 8: Location can be decoded better than chance across spatial cues. The population decoder was trained using responses of neurons to stimuli containing limited spatial cues, either ITDs alone (A-C), ILDs (D-F) or ILDs and spectral cues (G-I). Performance of the decoder was tested using the same neurons' responses to either the same stimuli (A, E & I) or stimuli containing different spatial cues, either ITDs (D, G), ILDs (B, H) or ILDs and spectral cues (C, F). The crosses represent performance when the decoder was trained and tested in the same paradigm for the trained stimulus in each case.

Out of pure curiosity, I wonder if the authors can make any statement how the decoding performances relate to the discrimination ability of spatial positions, given that the behavioural task required spatial discrimination (directionality, specifically).

This analysis is beyond the scope of the paper; relating discrimination to absolute localization abilities is not necessarily straightforward even within the behavioural domain (Moore, Tollin and Yin, Hearing Research, 2008). Understanding how neural responses relate to the perceptual elements of this task is ongoing work and, we hope, will form the basis of another manuscript at some point in the future.

Minor:

Line 284: Reference to panel E precedes C and D and then is repeated on line 291.

Thank you, this has now been rectified

Line 369: Two models are mentioned, but three are introduced (and shown in Fig. 7).

Apologies – we have edited this (see above response) and it now clearly describes all three models

Line 390: Potentially incorrect reference? 38 maybe?

Yes, it was reference 38, thank you for spotting this error!

Line 813: Surplus “.”

Deleted – thank you.

Line 897: What were the criteria to assign clusters as multi or single units?

Added in description:

“Units were defined as single if they contained fewer than 1% of all inter-spike intervals within 1 ms”

Line 1008: Reference to non-existent panel.

This should have referred to panel 7J and has been altered as such:

*For testing the number of channels with a fixed number of units (**Figure 7 J**), linear interpolation between the nearest actual values was performed so that the same, maximum number of units could be compared.*

Reviewers' comments:

Reviewer #1 (Remarks to the Author):

It would appear there are three excellent reviews of this manuscript and a general consensus among the reviewers that this is an outstanding contribution.

The authors have provided excellent edits and supplementary figures addressing my questions and those of other reviewers. The final manuscript is exciting and will no doubt have a huge impact on the field. I fully recommend acceptance and look forward to seeing the positive outcome in the scientific community!

Reviewer #2 (Remarks to the Author):

My opinion of this project remains high, and the quality of the manuscript has been appreciably improved by revision. I have a few specific comments intended to clarify the presentation.

Lines 6-7: This doesn't make sense: how could a neuron be sensitive to source location independent of the spatial cues for that location? I prefer the treatment on lines 36-37, which contrasts "specific localization cues" with "an integrated representation of space across cues".

Lines 10-12: I suggest that the observations of the present experiment be presented in past tense: cues had little impact; subpopulation of neurons encoded; decoders outperformed.

Lines 60-61: The numbers of sub-populations here is confusing. You say "conserved across a large sub-population" (i.e., one) then compare summed activity of two sub-populations. I think that you mean consistently "a small number of..." or "two or more" sub-populations.

It was challenging to discern the difference in the results between lines 180-194 compared with 204-211. After several readings, I figured out that one paragraph is counting percent of units showing significant decoding whereas the second was quantifying %MI among the significant units. Perhaps an effort could be made to draw that distinction more clearly.

Line 227: I suggest changing to "rather than individual [or specific] localization cues".

Line 231: The possessive form of it is "its", not "it's"

Line 565: The observation of a sharpening of spatial tuning by addition of a continuous competing sound also has been reported in cortical area A2 of anesthetized cats (Furukawa and Middlebrooks, J. Neurophysiol 86: 226-240, 2001), particularly in their "diffuse" condition and when a focal masker was fix at 0-degrees azimuth (as in the present study).

Line 775: Reference 55 (Zhou et al.) is incomplete.

Reviewer #3 (Remarks to the Author):

The authors made considerable additions and clarifications, which enhance the comprehensibility of the study. However, I am still concerned about the interpretation of some of the central findings. In particular, the responses to my previous points actually substantiated my concerns about the authors' conclusions that "A1 represents the location of auditory objects in a cue

invariant manner” and that “a labeled line model decoder outperforms two-channel model decoders”.

In regard to cue invariant representation of locations of auditory objects in A1:

- The authors themselves state that only a subpopulation (of 20% to 30%) of the recorded units in A1 is significantly responsive to multiple stimulus classes (i.e. cues). They did not test how or to what extent the observed tuning properties facilitate perceptual object formation. It might well be that the rather consistent tuning across different stimuli by this subpopulation is crucial for a spatially unified percept of a spectrally complex source, and I believe that the presented data is very interesting in this regard, but it remains speculative at this point. Thus, the title is an overstatement.

- More importantly, I am not convinced by the authors’ claim that this subpopulation of units with significant tuning for >1 condition represent clear evidence for cue invariance. The authors state that in Fig. 4D-L, the centroids of these units would be “almost exclusively along the line of equality, indicating that their tuning in each condition is consistent”. This simply is not true. For many of these scatter plots (especially for those comparing conditions with distinct binaural cues), rather strong deviations from the line of equality are observed (~ 40° in Figs. 4D,G,J) or the data simply take the shape of a cloud (Fig. 4H,K). That fact that a paired T-test does not yield very low p-values can be explained by the effects that these deviations go both ways and that the sample size was rather small (compare e.g. Fig. 6F,G, where significant deviations are reported, yet the deviances from the line of equality appear less pronounced than in Figs. 4D-L). Moreover, even small deviations can be highly informative for a multi-channel decoder, so it is important to describe these effects of stimulus condition in detail to be able to correctly interpret the model performances in the later parts of the manuscript (see also below).

In regard to labeled line model decoders outperforming two-channel model decoders

- I still believe that the “labeled line model” is a unnecessarily confusing description given what the authors try to show. The authors state in lines 57ff: “The labeled line model posits that heterogeneous spatial tuning exists, with different cells narrowly tuned to specific sound locations [...] across the entire azimuthal plane”. This prerequisite of narrow and specific tuning is not met by the presented data (which was very broadly tuned with consistently contralateral preference centered around 45°). Moreover, the explicit goal of the authors (stated in the Introduction) is to investigate neuronal representations of auditory object locations. Yet in the auditory community, the term “labeled line” is very tightly linked to the ongoing (and already terribly confusing) debate on the neuronal detection and representation of sound location cues in brainstem in midbrain, and will be associated by most readers with the Jeffress model idea of dedicated channels for each particular spatial position. Yet cortical representations during active localization are likely to be highly plastic (e.g. Lee and Middlebrooks 2011). Hence, given a particular context, the response of individual neurons to a specific spatial position might change, which is contradicting the labeled line idea. I am not arguing that a pattern decoder might be the correct model description underlying neuronal representation of spatial location during active localization in A1, but the term “labeled line” is misleading (see also my comment on cross validation below).

- I cannot follow the authors’ argument of their “matched channel analysis” representing evidence for the superiority of the labeled line model decoder. Since each unit represents a channel in the labeled line model, it benefits from the addition of more units, i.e. channels, particularly if variability exists in the tuning between units (which it does, Fig. 4). In contrast, the two-channel model averaged across units within channels, and thus does not take advantage of this variability, and thus shows inferior performance, even if the same number of neurons is provided.

- The authors state in their rebuttal about the shuffling analysis: “The key take-home message from this analysis is therefore that the benefit of a many-channeled system isn’t just that it has more channels, but the labels of these channels are critical for spatial decoding. If this were not true, then shuffled labelled line systems with many channels would perform better than two-channel models, simply as a bi-product of having more channels.” Yet to my understanding, shuffling unit sequence only shows that the tuning preference of all units within a particular channel must be similar for correct pattern identification. To me, this appears to be a trivial finding (and has been studied in some depth already, e.g. Luelig et al. 2011; Goodman et al., 2013), as

the averaging within channels simply diminishes the tuning of the channel if individual units are not similarly tuned (which is what the “labels of the channels” really is). This interpretation is corroborated by the report (lines 476ff) that the effect of shuffling unit order diminished for high number of channels, as only a few units per channel contribute.

- New Supplemental figure 8 shows that during cross validation, performance of the decoder drops, which implies that units are not “cue invariant” and that the changes in unit responses for the different stimuli do affect the decoding.
- If I am not mistaken, the Fig S8 only shows data from the modified two-channel model, while cross validation data from the labeled line decoder is not shown? Such a comparison of models is crucial to evaluate the authors’ central claim that one model is superior. Given the data in Fig 7H-K, one could assume that the model condition containing 50 channels corresponds to a “labeled line model”, in which case Fig S8 actually shows that there is no clear outperformance of one model over the other.
- Moreover, I maintain my assessment that the cross validation data is central to the authors’ claim of cue invariant representation of location, and thus this figure (plus the corresponding data for the labeled line model) should not be in the supplementary material.

Reviewers' comments:

Reviewer #1 (Remarks to the Author):

It would appear there are three excellent reviews of this manuscript and a general consensus among the reviewers that this is an outstanding contribution.

The authors have provided excellent edits and supplementary figures addressing my questions and those of other reviewers. The final manuscript is exciting and will no doubt have a huge impact on the field. I fully recommend acceptance and look forward to seeing the positive outcome in the scientific community!

Thanks!

Reviewer #2 (Remarks to the Author):

My opinion of this project remains high, and the quality of the manuscript has been appreciably improved by revision. I have a few specific comments intended to clarify the presentation.

Lines 6-7: This doesn't make sense: how could a neuron be sensitive to source location independent of the spatial cues for that location? I prefer the treatment on lines 36-37, which contrasts "specific localization cues" with "an integrated representation of space across cues".

Agreed, altered as suggested

Lines 6-7: Specifically, whether neurons in auditory cortex represent spatial cues or an integrated representation of auditory 'space' across cues is not known.

Lines 10-12: I suggest that the observations of the present experiment be presented in past tense: cues had little impact; subpopulation of neurons encoded; decoders outperformed.

Agreed and altered as suggested.

Lines 60-61: The numbers of sub-populations here is confusing. You say "conserved across a large sub-population" (i.e., one) then compare summed activity of two sub-populations. I think that you mean consistently "a small number of..." or "two or more" sub-populations.

Yes, thank you for this. Changed to 'two or more'

Lines 61-65: In contrast, the two-channel model^{38,39} posits that tuning of cells is broad and conserved across a small number of a large sub-populations of cells, with space represented by the relative summed activity of two or more sub-populations (defined either by the hemisphere of the brain in which cells were recorded, or the hemifield of space to which cells are primarily tuned).

It was challenging to discern the difference in the results between lines 180-194 compared with 204-211. After several readings, I figured out that one paragraph is counting percent of units showing significant decoding whereas the second was quantifying %MI among the significant units. Perhaps an effort could be made to draw that distinction more clearly.

Thank you for pointing this out – we hope that the following edits make the distinction more clearly

Lines 168-180:

We first used this measure to ask what proportion of units conveyed significant spatial information and found that 60% (153 / 253) of units were spatially informative in at least one temporal resolution. All three decoding windows resulted in a similar proportion of spatially informative responses (logistic regression of bin width vs. constant model, $X^2 = 2.88$, $p = 0.0896$, d.f. = 1, Figure 3H).

While a similar proportion of units were spatially informative across all bin widths, the advantage of this decoding approach is that we can quantify how informative individual responses are. We therefore considered decoding performance, expressed as the percentage of the maximum available MI for perfect classification performance (where the maximum was defined as $\log_2(\# \text{ locations})$, see methods) and observed that performance was highest with the 150 ms bin width (i.e. a spike rate code, Figure 3 I).

Line 227: I suggest changing to “rather than individual [or specific] localization cues”.

Changed as suggested.

Line 231: The possessive form of it is “its”, not “it’s”

Thank you, changed.

Line 565: The observation of a sharpening of spatial tuning by addition of a continuous competing sound also has been reported in cortical area A2 of anesthetized cats (Furukawa and Middlebrooks, J. Neurophysiol 86: 226-240, 2001), particularly in their “diffuse” condition and when a focal masker was fix at 0-degrees azimuth (as in the present study).

This reference has been added, thank you.

Line 775: Reference 55 (Zhou et al.) is incomplete.

Thank you for spotting this, the reference has been updated.

Reviewer #3 (Remarks to the Author):

The authors made considerable additions and clarifications, which enhance the comprehensibility of the study. However, I am still concerned about the interpretation of some of the central findings. In particular, the responses to my previous points actually substantiated my concerns about the authors’ conclusions that “A1 represents the location of auditory objects in a cue invariant manner” and that “a labeled line model decoder outperforms two-channel model decoders”.

We have performed additional analysis both at the level of individual units (to establish significance levels for whether e.g. centroid position changes significantly across stimulus types) and population decoders (implementing all of the reviewer’s suggestions). The results of these analysis substantiate our claim that a subpopulation of units represent space rather than specific localization cues. These additional analyses and the resulting changes to the manuscript are detailed below.

In regard to cue invariant representation of locations of auditory objects in A1:

- The authors themselves state that only a subpopulation (of 20% to 30%) of the recorded units in A1 is significantly responsive to multiple stimulus classes (i.e. cues). They did not test how or to what extent the observed tuning properties facilitate perceptual object formation. It might well be that the rather consistent tuning across different stimuli by this subpopulation is crucial for a

spatially unified percept of a spectrally complex source, and I believe that the presented data is very interesting in this regard, but it remains speculative at this point. Thus, the title is an overstatement.

We acknowledge that our data do not conclusively link this subpopulation to auditory object formation and have edited the title accordingly. We have therefore changed the title from “Primary auditory cortex represents the location of auditory objects in a cue-invariant manner” to “Neurons in primary auditory cortex represent sound source location in a cue invariant manner”.

• More importantly, I am not convinced by the authors’ claim that this subpopulation of units with significant tuning for >1 condition represent clear evidence for cue invariance.

As detailed further below, we have performed additional analysis of both single unit and population decoder performance to substantiate our claim that a subpopulation of cells represent sound source location rather than a specific localization cue (i.e. are cue invariant).

Our reasons for this claim are as follows (each point is further expanded in the response letter below to address specific points as raised by the reviewer):

- a population of single units are informative about auditory space across multiple conditions (Figure 5c).*
- for all but a handful of such units, the centroids measured in any pair of stimulus conditions are not significantly different when a within-unit significance test is applied (Figure 4, supplementary Fig.5).*
- when the responses of these jointly tuned units are used to build a distributed (/pattern / labeled line) decoder trained on the neural responses to ITD (low-pass) stimuli and tested on the responses to high-pass stimuli (no fine structure ITDs) performance is significantly better than chance (new Figure 8, supplementary Figure 9).*

The authors state that in Fig. 4D-L, the centroids of these units would be “almost exclusively along the line of equality, indicating that their tuning in each condition is consistent”. This simply is not true. For many of these scatter plots (especially for those comparing conditions with distinct binaural cues), rather strong deviations from the line of equality are observed (~ 40° in Figs. 4D,G,J) or the data simply take the shape of a cloud (Fig. 4H,K). That fact that a paired T-test does not yield very low p-values can be explained by the effects that these deviations go both ways and that the sample size was rather small (compare e.g. Fig. 6F,G, where significant deviations are reported, yet the deviances from the line of equality appear less pronounced than in Figs. 4D-L). Moreover, even small deviations can be highly informative for a multi-channel decoder, so it is important to describe these effects of stimulus condition in detail to be able to correctly interpret the model performances in the later parts of the manuscript (see also below).

We agree that there are a small number of units whose centroids (4 D, G, J) deviate by a large amount (but note that for a small proportion of units this is also seen when we perform repeated testing with the same stimulus, supplemental figure 6). Our claims about centroids do not relate to Figure 4H, K as these figures plot tuning width and modulation depth respectively.

We performed additional analysis that would enable us to determine whether individual units significantly changed their centroid (or modulation depth / tuning width) between stimulus conditions. To achieve this, we non-parametrically resampled the original data in order to estimate the 95% confidence limits on the centroid, ERRF and modulation depth for each unit. These results have been incorporated into figures 4 and 6 and supplementary figures 4, 5 and 6. They illustrate that very few units change, and we have edited the text to report this:

Line 221-227

In order to assess differences on an individual cell basis, non-parametric resampling of the data was performed to generate estimated confidence intervals on the centroid, ERRF width and modulation depth measurements (see methods). Using this measure, very few individual cells that were spatially modulated in pairs of cue-restricted conditions showed significant changes in centroid between the stimuli (open circles, Figure 4D-F, LPN-BPN: 3/22, LPN-HPN: 1/27, BPN-HPN: 2/23, Supplementary Figure 5).

Line 1029 (Methods):

To estimate confidence intervals for the centroid, ERRF width and modulation depth of each unit the spikes rates from each location were non-parametrically resampled with replacement 500 times, taking the same number of trials for each location. The centroid, ERRF width and modulation depth were subsequently calculated from the resampled data resulting in a distribution of 500 observations for each parameter. The centroid, ERRF width and modulation depth of each unit were then taken as the mean of each distribution with the 2.5th and 97.5th percentiles giving the confidence intervals for each measure.

A new supplementary figure that shows the confidence intervals of all the units in Figure 4 has been included to illustrate the extent of the intervals, which are generally quite large (Supplementary Fig. 5). Large confidence intervals could be due to technical or analytical factors, such as recording noise, or constructing the spatial receptive field from a limited numbers of stimulus repetitions (with that number constrained by the number of trials an animal will perform in the behavioural paradigm). The “noise” we observe could also reflect variability in firing rates of neurons that emerge due to other biologically relevant processes over which we do not have experimental control. Nonetheless, the cue-invariance of these units is supported by the cross-cue population decoding approaches we outline below.

In regard to labeled line model decoders outperforming two-channel model decoders

- I still believe that the “labeled line model” is a unnecessarily confusing description given what the authors try to show. The authors state in lines 57ff: “The labeled line model posits that heterogeneous spatial tuning exists, with different cells narrowly tuned to specific sound locations [...] across the entire azimuthal plane”. This prerequisite of narrow and specific tuning is not met by the presented data (which was very broadly tuned with consistently contralateral preference centered around 45°). Moreover, the explicit goal of the authors (stated in the Introduction) is to investigate neuronal representations of auditory object locations. Yet in the auditory community, the term “labeled line” is very tightly linked to the ongoing (and already terribly confusing) debate on the neuronal detection and representation of sound location cues in brainstem in midbrain, and will be associated by most readers with the Jeffress model idea of dedicated channels for each particular spatial position. Yet cortical representations during active localization are likely to be highly plastic (e.g Lee and Middlebrooks 2011). Hence, given a particular context, the response of individual neurons to a specific spatial position might change,

which is contradicting the labeled line idea. I am not arguing that a pattern decoder might be the correct model description underlying neuronal representation of spatial location during active localization in A1, but the term “labeled line” is misleading (see also my comment on cross validation below).

*We have changed the wording throughout the paper to reflect the reviewer’s concerns and have referred to this model as the **distributed** code. Further references have been added to clarify the terms of the models where people have used similar decoders with various nomenclature previously.*

- I cannot follow the authors’ argument of their “matched channel analysis” representing evidence for the superiority of the labeled line model decoder. Since each unit represents a channel in the labeled line model, it benefits from the addition of more units, i.e. channels, particularly if variability exists in the tuning between units (which it does, Fig. 4). In contrast, the two-channel model averaged across units within channels, and thus does not take advantage of this variability, and thus shows inferior performance, even if the same number of neurons is provided.

The reviewer’s summary of this analysis distills the implication of our analysis nicely, that the distributed code takes advantage of neural variability that the two-channel model ignores.

The shortcoming of the two channel model in this context is intuitive from an analytical perspective, when one has access to many varied signals such as single and multiunit recordings we present here. However, we wanted to include the comparison of distributed and two channel models because of the latter’s prominence in auditory neuroscience. Neuroimaging studies in particular have been strongly influenced by the two channel model, yet the biological insights provided by these approaches have been technically constrained by the averaging of large-scale signals across many thousands of neurons. We believe it is thus important for us to make the point that such models perform poorly when we have access to the responses of individual neurons, as would be the case for any brain region that uses auditory cortical information in perception and decision making. Its also worth noting that not all previous studies determine an advantage for distributed coding – for example in gerbil IC both codes perform equivalently (Belliveau et al., 2014).

Finally, we go on to look at how population decoding varies as a function of channel number and within-channel population size (Figure 7H-K). A central motivation of this analysis was to better understand how averaging across neural populations impairs recovery of spatial information and thus how the auditory system might leverage variability in activity across neurons.

- The authors state in their rebuttal about the shuffling analysis: “The key take-home message from this analysis is therefore that the benefit of a many-channeled system isn’t just that it has more channels, but the labels of these channels are critical for spatial decoding. If this were not true, then shuffled labelled line systems with many channels would perform better than two-channel models, simply as a bi-product of having more channels.” Yet to my understanding, shuffling unit sequence only shows that the tuning preference of all units within a particular channel must be similar for correct pattern identification. To me, this appears to be a trivial finding (and has been studied in some depth already, e.g. Luelig et al. 2011; Goodman et al., 2013), as the averaging within channels simply diminishes the tuning of the channel if individual units are not similarly tuned (which is what the “labels of the channels” really is). This interpretation is corroborated by the report (lines 476ff) that the effect of shuffling unit order diminished for high number of channels, as only a few units per channel contribute.

Again, the reviewer's understanding of our analysis methods and interpretation is correct – however we would argue that while the effect of shuffling isn't necessarily novel by itself, it remains a critical validation of our decoding procedure. As the reviewer states, spatial tuning of the channels reflects what the channel label really is. We wanted to make sure this was true and the presentation of this data is thus a vital control. Furthermore, the fact that multiple studies have gone into detail in investigating such issues reflects the importance of the analysis.

- New Supplemental figure 8 shows that during cross validation, performance of the decoder drops, which implies that units are not “cue invariant” and that the changes in unit responses for the different stimuli do affect the decoding. If I am not mistaken, the Fig S8 only shows data from the modified two-channel model, while cross validation data from the labeled line decoder is not shown? Such a comparison of models is crucial to evaluate the authors' central claim that one model is superior. Given the data in Fig 7H-K, one could assume that the model condition containing 50 channels corresponds to a “labeled line model”, in which case Fig S8 actually shows that there is no clear outperformance of one model over the other.

Moreover, I maintain my assessment that the cross validation data is central to the authors' claim of cue invariant representation of location, and thus this figure (plus the corresponding data for the labeled line model) should not be in the supplementary material.

We were initially reluctant to perform this analysis as our sample size for the analysis is really small. When requested by the reviewer in the first round of revisions we used simulated data to attempt to address the problem (and which, as the reviewer notes, yielded somewhat unclear answers). However, we are grateful that the reviewer pushed us to return to this analysis as we think that the resulting conclusions considerably strengthen our manuscript. We also note that the reviewer is right in pointing out this analysis should be performed with the distributed decoder rather than the two-channel decoder, as the distributed decoder yields superior performance. We therefore used the distributed decoder on the subpopulation of units that were spatially informative across pairs of stimulus conditions (i.e. the subpopulations we claim represent space rather than localization cues). For the main paper (new Figure 8, see also supplemental figure 9) we focused on the comparisons of high-pass and low-pass stimuli as these stimuli restrict the available localization cues to contrasting binaural cues (fine structure ITDs are eliminated in the high-pass case, and only ITDs are available in the low-pass case). In this analysis a decoder trained on high-pass stimuli was able to recover sound location from neural responses to low-pass stimuli (on which it had not been trained) and vice versa. As in Higgins et al.'s (2018, PNAS) functional imaging study, we quantify the decoder performance both in terms of the magnitude of the errors and the % correct performance. Like Higgins et al., we observe that the cross-condition decoder yields performance that is worse than within-condition decoding, but significantly better than chance. Given the very small number of units available for this analysis (n=16) we think this, coupled with the observation that these units maintain their spatial tuning and centroid positions, is compelling evidence that this subpopulation of units does indeed represent space rather than a single type of cue. It seems likely that if more units were available for this analysis, we would be able to recover sound location with cross-cue decoders with as much precision as within-cue decoders.

The relevant sections of text from the manuscript are:

Lines 465-493 (Results): Finally, if neurons represent spatial location, rather than merely spatial cue values, it should be possible to train a decoder with the responses to one class of stimuli and recover sound location when tested with the responses to a different class of stimuli. To test this, the

normalised (z-scored) responses of neurons with significant spatial information (MI) in pairs of stimulus conditions were used to train and test the distributed model decoder across stimulus types. We trained the distributed decoder with responses to one condition (e.g. LPN) and tested the decoder using responses from the same neurons to another condition (e.g. HPN). To assess cue-invariance we chose the conditions that differed most clearly in the available binaural cues, the low-pass and high-pass stimuli, which contained only ITDs, or eliminated fine-structure ITDs to leave ILDs and spectral cues, respectively (Figure 8). To quantify decoder performance we considered both the % correct score and the unsigned error magnitude (mean observed rms error, RMSE) and compared these values to that observed when the identity of the cells was shuffled prior to decoding (Figure 8B, RMSE was considered significantly lower than chance if the observed mean RMSE was less than the 5th percentile of the shuffled distribution). While decoding stimulus location with a decoder trained on the cross-cue condition resulted in worse performance than the within-cue condition (Figure 8A), the error magnitudes were smaller than that expected by chance (Figure 8B). For decoders trained on LPN and tested on HPN performance was higher than that expected by chance (Figure 8C), with performance close to exceeding the chance level ($p = 0.051$) for a decoder trained on HPN and LPN.

A similar comparison of low-pass (ITD) and band-pass stimuli (narrowband ILD) was performed (Supplementary Figure 9). Cross-trained decoders showed smaller than chance RMSE when trained with BPN and tested with LPN. Interpreting the lack of generalisation from LPN to BPN is difficult: the decoding is done on a very small number of units, and the narrowband nature of the band-pass condition render it highly unnatural and could explain why this manipulation had the largest effect on performance of the ferrets in the relative localisation task (Figure 1D).

Lines 511-532 (Discussion):

Ferrets were able to perform the relative localization task using either ILD or ITD cues as evidenced by performance when cues were restricted to ITDs (low-pass noise), mainly ILDs (band-pass noise) or ILDs and spectral cues (high-pass noise). In such conditions, the directional preference of cells (as measured by centroids) was stable, and we identified a subset of roughly a quarter of units that were significantly spatially informative when provided with either ILDs or ITDs. Moreover, decoders trained on the neural responses to low-pass stimuli were able to recover location from the responses to high-pass stimuli and vice versa (Figure 8). Together, these results suggest that a subpopulation of A1 cells provide a cue-invariant representation of sound location. The population of neurons that we were able to record across multiple stimuli was small (we recorded the responses of 16 spatially modulated units in both high-pass and low-pass conditions). While these small populations represented sound location with near-perfect accuracy when trained and tested with the same stimulus type, performance declined in the cross-cue condition. With a larger population of jointly-sensitive neurons it seems likely that performance could be maintained across multiple stimulus types containing contrasting localisation cues. Other units only conveyed spatial information in the presence of ITDs or ILDs. Similar results have been obtained using human neuroimaging where regions of cue-independent and cue-specific voxels were observed²³. These neuroimaging results suggest that a representation of azimuthal space exists within auditory cortex that is independent of its underlying acoustic cues. Our results provide the first cellular resolution evidence from the auditory cortex of behaving subjects in support of this hypothesis.

REVIEWERS' COMMENTS:

Reviewer #3 (Remarks to the Author):

The authors made considerable improvements to the manuscript, especially by providing more detailed information regarding the variability of tuning properties across stimulus conditions and their significance. I also appreciate the new information about the model performance during cross validation. My impression persists that comparison of model performances is not adding to the paper, but I concede that this might be a subjective bias.

I only have two remaining comments:

- 1) The authors show in Supplementary Fig. 6 that sizable changes in tuning properties exist across days. I wonder how this variability would affect decoding performances for the different models, especially during cross validation (LPN to HPN etc).
- 2) The legend to Supplementary Fig. 5 does not correspond to the data provided in the panels. Particularly, I am missing the illustration of confidence intervals by crosshairs (they seems to be only provided for units with significantly different tuning). Additionally, labels are missing for most panels.

REVIEWERS' COMMENTS:

Reviewer #3 (Remarks to the Author):

The authors made considerable improvements to the manuscript, especially by providing more detailed information regarding the variability of tuning properties across stimulus conditions and their significance. I also appreciate the new information about the model performance during cross validation. My impression persists that comparison of model performances is not adding to the paper, but I concede that this might be a subjective bias.

I only have two remaining comments:

1. The authors show in Supplementary Fig. 6 that sizable changes in tuning properties exist across days. I wonder how this variability would affect decoding performances for the different models, especially during cross validation (LPN to HPN etc).
 - a. *We have added in a new supplementary figure (Supplementary figure 9) which explicitly tests this: it shows the performance of the distributed decoder when trained and tested on neural activity recorded in response to the same stimuli, from the same neurons, but recorded on different days. These data show that the drop in % correct is similar to that observed when training and testing results from the same neurons to stimuli containing different localization cues.*
2. The legend to Supplementary Fig. 5 does not correspond to the data provided in the panels. Particularly, I am missing the illustration of confidence intervals by crosshairs (they seems to be only provided for units with significantly different tuning). Additionally, labels are missing for most panels.
 - a. *Apologies, the panels now correspond to the figures.*